# The Colombian scientific elite—Science mapping and a comparison with Nobel Prize laureates using a composite citation indicator

**Julián D. Cortés**[1,2,3]*, **Daniel A. Andrade**[4]

**1** School of Management and Business, Universidad del Rosario, Bogotá, Colombia, **2** Fudan Development Institute, Fudan University, Shanghai, China, **3** School of Business, Woxsen University, Sadasivpet, Telangana, India, **4** Independent, Bogotá, Colombia

* julian.cortess@urosario.edu.co

**Data Availability Statement:** Data available in the permanent link that follows: https://doi.org/10.34848/GJO6SY Datasets include: List of the AAEP awarded 1990-2020 (xlsx format): lead author

## Abstract

A well-established agenda on the research output, impact, and structure of global scientific elites such as Nobel Prize laureates has generated interest in the scientific elites from developing countries. However, this topic has not been investigated in detail. This study, first, deploys science mapping techniques to provide a comprehensive analysis of the output, impact, and structure of the Colombian scientific elite, i.e., researchers awarded with the *Alejandro Ángel Escobar Foundation National Prize* 1990–2020, known locally as the *Colombian Nobel*. Second, we conducted a productivity and impact comparison between the Colombian scientific elite and Nobel Prize laureates in science and economics by means of a stratified random sample 1990–2020 via the composite citation indicator proposed by Ioannidis et al. Findings showed that the Colombian scientific elite has a broader agenda than indexing titles in internationally renowned bibliographic databases. The Colombian scientific elite also showed positive growth, which is an inverse trend compared with the sample of Nobel laureate productivity. There were no noticeable changes in productivity/impact before and after receiving the *Alejandro Ángel Escobar Foundation National Prize*. Institutional collaboration within the Colombian scientific elite displayed the highest betweenness (*brokerage*) role of world/local top-tier universities. However, only two Colombian scientific elite members published an article with two Nobel Prize laureates. Most of the research profiles reflected the national output priorities, but were found to diverge from the national focus in respect of strategic research capacities. The interleaving of the Colombian scientific elite and Nobel Prize laureates—particularly between the 3rd and 2nd quartiles—enabled a more nuanced analysis of the local impact in the global scientific landscape. Our findings also contrast with previous findings on the lower research impact of authors from Latin America, despite their involvement as contributors to reputable journals, and also shed light on the research performance-impact standards and agenda between the global North and South and provide an in-context assessment of outstanding local research.

awarded-AAEP; author sex; year; award category; document title; abstract (Spanish); affiliation at the time the award was received; institution's country and city; highest/last degree; and the university that granted the academic degree and country. Processed matrices for the following science mapping networks (csv format): Coauthorship matrices at the institutional and author levels by AAEP category: six matrices; bibliographic coupling matrices by AAEP category: three matrices.

**Funding:** JDC Seed fund Universidad del Rosario https://www.urosario.edu.co/ The funders had no role in study design, data collection and analysis, decision to publish, or preparation of the manuscript.

**Competing interests:** The authors have declared that no competing interests exist.

## Introduction

On a humorous note, Richard J. Roberts—Nobel Prize winner in physiology/medicine—outlined ten *simple* rules to win a Nobel Prize and be part of the global scientific elite (GSE) [1]. Among these rules were the following: *work in the laboratory of a previous Nobel Prize winner; try to work in the laboratory of a future Nobel Prize winner*; or *pick your family (i.e., Nobel laureates) carefully*. For developing countries such as Colombia, none of those rules are *simple* considering the null population of Nobel laureates in science currently teaching/researching at a national university.

GSEs push the frontiers of knowledge. Yet, despite a well-established agenda [2–27], researchers or scientific awardees from developing countries have been sidelined. Let us consider two well-known examples: Nobel Prize laureates and highly-cited and productive institutions/researchers [9, 10, 28, 29]. First, seventy-seven percent of the Nobel Prize laureates in physics had US, German, UK, French, or Russian citizenship [30]. In contrast, 2% had citizenships from developing countries such as China, Pakistan, India, or Morocco [30]. Second, most of the world's scientific wealth (i.e., research output and citations) has been accumulated in a few premier institutions in developed countries [31–33]. Such staggering inequality is reflected in the fact that the top 1% of most-cited authors constitutes over a fifth of all citations globally [34]. To understand why this is the case, it is necessary to look at the nature of scientific elites through the lens of developing countries' historical, economic, institutional, and cultural contexts, research standards, and affiliations. [35–40].

This study focuses on Colombia, which ranks among the top five countries in Latin America in total document output and the top-fifty in total citations worldwide 1996–2019 [41]. Colombia's national investment in science, technology and innovation activities is a mere 0.8% of the GDP (gross domestic product) (2015–2020) [42]. Similarly, the country's investment in R&D (research and development) has yet to surpass 0.4% of the GDP (2015–2020), which is below the Latin American average of 0.7% (2013) [42, 43]. The Latin American region is also below the world average of 2.2% in R&D investment (2018) [44]. Regarding the scientific workforce, Colombia has 58 researchers per million inhabitants (male: 65%; female: 35%), compared with 260 in Mexico (male: 67%; female: 33%) or 1,206 in Argentina (male: 47%; female: 53%) [43]. At the other end of the spectrum, i.e., in the high-income countries, Denmark has 7,310 researchers per million inhabitants (male: 65%; female: 35%) and Finland 7,009 (male: 68%; female: 32%) [43]. This disparity is all the more striking when viewed in the context of Noble Prize laureates from Colombia, the subject of this study.

The Nobel Prize has been awarded to Gabriel García Márquez (literature) and Juan Manuel Santos (peace), but none in the science categories. Likewise, only one Colombian researcher is listed in the 2020 edition of Clarivate's Highly Cited Researchers: Olga Sarmiento, Universidad de Los Andes [30, 45]. Based on these standards, a Colombian scientific elite (CSE) is non-existent. Colombia does, however, have its own *Nobel Prize* equivalent: the *Alejandro Ángel Escobar Foundation National Prize* (AAEP) [46, 47]. The organization's founder—inspired by the Swedish inventor and his legacy—stated that the AAEPs "*are to be awarded for truly meritorious work that deserves the mark of excellence at least within the cultural context of the country.*" [48]. Each year the AAEF invites all researchers of Colombian nationality—regardless of local or international affiliation—to submit their research for assessment by the Foundation's committee (i.e., peer-reviewed articles or books, master's or Ph.D. thesis, technical reports, independent research). If the document is multiauthored, the authors have to assign a representative/coordinator, who will receive the prize money and a silver medal. The representative/coordinator must be a native Colombian. This national *recognition* has been awarded annually since 1955. There are three science categories [48]: 1) *physics and natural sciences*; 2)

*social sciences and humanities*; and 3) *environmental sciences and sustainable development*. There is an *honorable mention* for each category if the jury decides so [48] and these acknowledged researchers are considered members of the CSE (e.g., Salomón Hakim [neurosurgeon] in 1967 and 1974; and Ana María Rey [physicist] in 2007).

The purpose of studying the CSE — and scientific elites in developing countries in general — is to understand its performance and research and collaboration structures in comparison with the GSE. The findings shed light on the research performance-impact standards and agenda between the global North and South. They also provide an in-context assessment of outstanding local research [49–52] amid the decreasing share of new Nobel Prize laureates from North America and an inverse trend from the Asia-Pacific region [4]. To the best of our knowledge, no research has been conducted on the CSE or any other developing country's scientific elite using bibliometric techniques [53]. Accordingly, this study aims to provide a comprehensive analysis of the output, impact, and structure of the CSE and draw a comparison with the GSE. This inquiry is guided by the following research questions (RQs):

- RQ1: Is the CSE more productive/cited before or after receiving the AAEP? [6, 11, 22, 54, 55]

- RQ2: Does the CSE collaboration network have any participation in the GSE or world top-tier institutions? [26, 33, 56]

- RQ3: What are the research fronts of the CSE? [7], and

- RQ4: Is the CSE *light years* away from the productivity and impact of the GSE? [31, 36, 57–59]

Following this introduction, section 2 reviews recent literature on GSEs. Section 3 outlines the data, methods, and techniques implemented. The methods and techniques implemented are coauthorship networks, both at the institutional and author levels; bibliographic coupling; and a comparative sample of 82 researchers (i.e., 41 AAEPs and 41 Nobel Prize laureates) using the composite citation indicator proposed by Ioannidis et al. [55]. Coauthorship networks map scientific collaboration, enabling researchers to identify the extent to which social connections contribute to the co-creation of knowledge. Bibliographic coupling discloses the clustering of research fronts, while the proposed composite citation indicator addresses total impact, normalized coauthorship, and author order based on six indicators at the author level. Section 4 presents the results to be discussed in section 5. Finally, section 6 presents the conclusions, limitations, and future agenda.

## Literature review

Research on the GSE is well-established in the informetrics literature [2–15, 21–24, 26, 27, 60–63]. For example, a Boolean search on Scopus's bibliographic database with the keyword "Nobel Prize" limited to 14 core journals on informetrics and research evaluation (e.g., *Journal of Informetrics*, *Scientometrics*, *JASIS&T*) [57], returned 75 results. We limited the review to three areas of interest: 1) international awards/prizes networks; 2) latest research on production and impact of the GSE; 3) and on the intellectual and social structure of the GSE.

First, Zheng and Liu [18] developed a "co-awardees" network of significant international awards/prizes according to awardees' assessment, establishing a similarity between the Nobel Prize and other awards/prizes awardees (e.g., Wolf, Lorentz, and Shaw awards/prizes). Subsequent work by Ma and Uzzi [13] assembled a scientific prize network based on 3,000+ prizes and the careers of 10,000+ prizewinners over a 100-year period. They found that the number of prizes doubled every 25 years, that the science hierarchy is becoming more vertical, and that having an awarded advisor is essential for winning at least one prize. The winning of subsequent prizes, however, is more a matter of expanding one's own network.

Second, recent work on the GSE concluded that the business of *predicting the next Nobel* had become a fruitless exercise since the laureates rank among the 500 most-cited authors after the 1970s compared to those awarded in the early twentieth century and their contributions have been limited to research niches rather than the discipline as a whole [22]. This was further refined by Ioannidis et al. [7], who found that of the 114 domains investigated, only five (i.e., particle physics [14%], cell biology [12.1%], atomic physics [10.9%], neuroscience [10.1%], and molecular chemistry [5.3%]) accounted for 52.4% of the Nobel Prizes awarded. Taking a closer look at the features of the Nobelists' papers, Zhou et al. [54] estimated that 74.7% were cited more than 500 times, innovative research was more cited than theoretical and experimental methods, and most of the papers were published in journals with an Impact Factor between 5–10. Ioannidis et al. [15, 55] examined the most-cited biomedical researchers and proposed a composite citation indicator assessment of 84,000+ highly-cited scientists in 12 fields. Among their findings, a number of scientists stated that research related to progressive evolution (i.e., continuous progress; broader interest; greater synthesis) rather than revolution was the most characteristic feature of a blockbuster paper. Based on the proposed composite citation indicator, many Nobelists ranked among the top-1000 highly-cited authors, but would rank much lower if based solely on citations. Similarly, highly-cited authors had published either none or very few influential papers as a first, single, or last author.

One of the most comprehensive studies on Nobelists' careers [11] found that they were productive from the outset (twice as many papers as a random scientist), showed a six-fold increase in publishing *hit* papers (top 1% of rescaled 10-year citations) and published on average two hit papers. Nevertheless, the overall career path before winning the prize is similar to that of other scientists. While the laureates' collaboration network did not increase after the Nobel Prize, it tended to be more consistent in productivity and impact. Contrasting Nobelists with highly-cited researchers, Kosmulski [6] — the only study that included the *Sveriges Riksbank Prize in Economic Sciences in Memory of Alfred Nobel* — argued that the virtue of publishing *hot papers* (0.1% of the top papers in the field in the past two years) is not common among recent Nobel laureates while the number of scientists who have at least one highly-cited paper substantially exceeds 100,000. Kosmulski [6] also found that Nobel Prize papers connect topically diverse clusters of research papers [8]. Schlagberger et al. [33] highlighted US dominance at the national and institutional level given that four US institutions hold most of the Nobelists in physics, chemistry, and physiology (i.e., UC Berkeley, Columbia, MIT, and Princeton). While Nobelists are mobile, they generally hail from the US, UK, Japan, and Germany.

Third, studies on the Nobelist collaboration network [26, 64] found that laureates published fewer papers, but with a higher than average citation, a feature further supported by Li et al. [11]. Nobelists tend to play a *brokerage* role in the collaboration networks by building intellectual and social bridges and exploiting *structural holes*. Jiang and Liu [64], for example, noted a high level of institutional inequality across periods. The most connected institution during 1990–1940 was the Humboldt University; during 1941–1980 it was the University of Cambridge, and during 1981–2017 it was Harvard University, a phenomenon outlined earlier [33].

Despite the considerable research already undertaken, the above literature review exposes two major limitations. First, as noted by Zheng and Liu [18] and Ioannidis et al. [7], further studies are needed that will incorporate other disciplines such as the social sciences and humanities. Second, since the study by Schlagberger et al. [33] focused solely on US institutions, a broader landscape is needed to research institutions in developing countries. To close these two gaps, this study includes researchers in social sciences and humanities, environmental science and sustainable development, who for the most part are affiliated with Colombian institutions. While the latter are not comparable to US institutions in terms of global

reputation, some of these Colombian institutions have garnered a regional reputation, particularly in Latin America and the Caribbean (e.g., Universidad Nacional or Universidad de Los Andes) [65].

## Materials and methods

### Data

**Colombian scientific elite.** The CSE list was sourced from the AAEF website (2000–2020) and from a book published by the AAEF (1990–1999) commemorating its half-century [53]. We decided to restrict our sample to the last 30 years' awardees in light of Colombian researchers' late involvement in publishing research articles in international journals (since the early 1990, ~200 papers were published annually in the Science Citation Index) [66]. We further restricted our sample to the leading author or representative in the case of multi-authored documents. Table 1 presents the CSE sample of 87 awardees categorized by sex and award. Female researchers have a 25.3% participation among the awardees, with the highest participation in social sciences and humanities (SoSci) with 11.4%. In contrast, male researchers have 74.7% of the total participation, the highest being in physics and natural sciences (PhySci) with 32.2%. While the issue of sex differential among scientists lies beyond the scope of this study, it is worth noting that the differential among AAEPs is similar to that among Nobel Prize winners. Evidence shows [67] that ten women were awarded a Nobel Prize in the sciences between 2004–2019, which is the same number of awardees during the first 100 years of Nobel history. Multiple causes have been cited to account for this discrepancy, such as marital and maternity status, lack of role models, and a lack of interest in following an academic career. All of which impact both productivity and further research for women.

S1 Table presents the higher/last academic degree by university and country according to the Colombian Ministry of Science, Technology, and Innovation platform for researchers' curriculum: CvLAC [68]. Thirty-one percent of researchers have completed an academic degree in the US and 28% a Ph.D. in reputable universities such as Harvard, MIT, Yale, or Wisconsin-Madison. In contrast, 27.6% have completed an academic degree in Colombia and 12.6% a Ph.D. in reputable Colombian universities such as Nacional, Antioquia, Valle, or Los Andes [68] (See also: Data availability).

**Bibliographic data.** Scopus was chosen over Web of Science (WoS) due to its broader journal coverage and researcher participation from developing countries, particularly Colombia [40, 69–71]. Based on the CSE list above, we searched and sourced the complete profiles in

**Table 1. Colombian scientific elite by sex and AAEF award category.**

| Total | 87 |
|---|---|
| **CSE sex by category** | % |
| **Female** | 25.3 |
| Environmental sciences and sustainable development–EnvSci | 8.0 |
| Social sciences and humanities–SoSci | 11.5 |
| Physics and natural sciences–PhySci | 5.7 |
| **Male** | 74.7 |
| Environmental sciences and sustainable development–EnvSci | 24.1 |
| Social sciences and humanities–SoSci | 18.4 |
| Physics and natural sciences–PhySci | 32.2 |

Source: [48, 53].

Scopus for each author. Those with only one indexed article were excluded. We also checked each author's current or past affiliation in the CvLAC [68] to avoid the inclusion of homonymous authors. The working sample of CSE with Scopus profiles consisted of 41 researchers, ~47% of the preliminary CSE list displayed in Table 1. For multiple-year AAEP awardees, the first year was considered for related assessments. Germán Poveda is a particular case in point, having been awarded three times (i.e., PhySci: 1999; EnvSci: 2007, 2019). We assigned him to the EnvSci category since he was awarded twice in that category. The CSE sample contains a similar sample of researchers compared to previous studies on non-Nobel laureates (e.g., 29 recipients of the *Derek John de Solla Prize Medal* [72]).

Table 2 presents the bibliometric descriptives. Environmental sciences and sustainable development (EnvSci) and physical sciences (PhySci) were the categories with the most profiles found in Scopus. PhySci was the category with the highest number of articles, authors, and citations per document. EnvSci, however, showed the highest annual growth rate. The most relevant periodical (most frequent) for SoSci was *Revista de Estudios Sociales* (Colombia–U. Andes), for EnvSci *Biotropica* (Wiley), and for PhySci *Physical Review A–Atomic Molecular and Optical Physics* (American Physical Society). This is consistent with the publishing and citation dynamics of the above disciplinary categories: SoSci is oriented towards the publication of books/book chapters in local journals, while PhySci-related disciplines tend to publish international research articles or conference proceedings [73–76]. Articles (921) with 10 + authors [77] were excluded from the analysis. These were mostly from medicine: 26%; earth and planetary sciences: 22.6%; and biochemistry, genetics and molecular biology: 18.6%. Such publications are difficult to assess since they could be either a product of publishing agreements or highly collaborative authors with marginal research input [59]. Due to Scopus' indexing accuracy, we only analyzed articles published between 1996–2020 [69].

## Methods

**AAEP research topics.** A semantic network was built based on each title of the awarded research document by category. The aim was to explore the document titles and examine the shared meaning and interconnection between key terms among titles [80]. A co-occurrence matrix was assembled to compute the number and direction of unique-word co-occurrence

**Table 2. Scopus author profiles and bibliometric descriptive of the CSE and GSE aggregated and by category 1996–2020.**

| Award category | Profiles | Documents | Articles | Annual growth % | Authors | Citations per article | Most frequent journal |
|---|---|---|---|---|---|---|---|
| *Alejandro Aangel Escobar National Prize—CSE* | | | | | | | |
| PhySci | 19 | 1,206 | 1,025 | 2.25 | 1,776 | 55.1 | *Physical Review A—Atomic Molecular and Optical Physics* |
| EnvSci | 14 | 485 | 439 | 6.67 | 735 | 26.4 | *Biotropica* |
| SoSci | 10 | 38 | 31 | <1 | 21 | 4.2 | *Revista de Estudios Sociales* |
| **Total** | **41** | **1,731** | **1,195** | **2.97** | **2,532** | **28.6** | |
| *Nobel Prize—GSE* | | | | | | | |
| Chemistry | 13 | 2,944 | 2,6 | -2.71 | 3,660 | 126.2 | *Journal of the American Chemical Society* |
| Economics | 6 | 334 | 226 | -0.39 | 247 | 98.2 | *American Economic Review* |
| Physics | 12 | 2,517 | 2,166 | -0.31 | 2,094 | 61.9 | *Japanese Journal of Applied Physics* |
| Physiology or Medicine | 10 | 1,211 | 897 | -5.16 | 2,558 | 113.2 | *Proceedings of the National Academy of Sciences of the United States of America* |
| **Total** | **41** | **7,006** | **5,889** | **-2.14** | **8,559** | **99.9** | |

Source: [48, 53, 68, 78, 79]. Abbreviations: EnvSci: environmental sciences and sustainable development; SoSci: social sciences and humanities; PhySci: physics and natural sciences.

after removing stop/period and non-informative words. Finally, a directed-weighted semantic network was produced based on the co-occurrence matrix. Two network analysis characteristics/metrics were computed: community detection; and betweenness. For the former, we implemented the Blondel et al. [81] modularity appraisal algorithm, which gives a node's capacity for mediating the flow of information between multiple clusters [82]. The equation for the betweenness calculation is:

$$C_B(p_k) = \sum_{i<j}^{n} \frac{g_{ij}(p_k)}{g_{ij}}; i \neq j \neq k \tag{1}$$

Source: [83].

where $g_{ij}$ is the shorter path that links nodes $p_i$ and $g_{ij}(p_k)$ is the shorter path that links nodes $p_i$ and $p_j p_k$. The higher the value, the higher its betweenness.

**Output and citations.** We outlined a descriptive section on articles and citations by category. We also explored the annual citation per article of the top three most-cited researchers per category before and after receiving the AAEP.

**Institutional collaboration and coauthorship.** Bibliographic data of the CSE profiles was processed with bibliometrix for R [79, 84]. Once this data has been processed and converted into the corresponding objects, preprocessing and cleaning are carried out in order to unify the authors' names and affiliations.

We used a multi-level approach for all the science mapping techniques implemented (i.e., coauthorship and bibliographic coupling analyses). A multi-level approach means analyzing the different structural features of a network at the macro (i.e., density and average path length); meso (i.e., community detection); and micro (i.e., betweenness centrality) levels through the use of network indicators [85]. Regarding the multi-level network indicators used, density indicates the degree to which both authors and institutions are connected (i.e., the number of connections that *exist* compared to the number of connections that *could* exist). The equation for the density calculation is:

$$den = 2L/n(n-1) \tag{2}$$

Source: [86].

where $L$ is the number of links and $n$ is the number of nodes. The average path length computes the average number of steps along the shortest path for every pair of nodes (i.e., authors, institutions) in a given network. The equation for the average path length calculation is:

$$l_G = \frac{1}{n(n-1)} \sum_{i \neq j} d(v_i, v_j) \tag{3}$$

Source: [87].

where $d(v_i, v_j)$ is the length of the shortest path that exists between two nodes. We used the Leiden algorithm to identify communities (i.e., clusters) [88]. Betweenness equation and interpretation were given in Eq (1).

**Bibliographic coupling.** A bibliographic coupling connects two documents if a common item was cited and appeared within both lists of references [89]. This facilitates an analysis of the clustering of shared items between documents and an investigation of disciplinary fronts that highlights how academic knowledge is shared [90–92]. This technique moderately outperforms other science mapping techniques used to identify research fronts (i.e., co-citation analysis, direct citation) [93]. It is also an extremely serviceable tool in the analysis of highly interdisciplinary fields and has been used to analyze SDGs (Sustainable Development Goals) [94], and the complete set of *Nature* journal publications spanning 150 years, ranging from the

arts and social sciences to earth and space and clinical medicine. It is, therefore, an appropriate method for analyzing such a highly interdisciplinary category as the PhySci category of the AAEP [95].

The equation for obtaining a bibliographic coupling network is:

$$B_{coup} = AxA'$$  (4)

Source: [79].

where $A$ is a document and $x$ is a cited reference matrix. The element $b_{ij}$ indicates how many bibliographic couplings exist between documents $i$ and $j$. $B_{coup}$ is both a non-negative/symmetrical matrix. The number of shared references defines the strength of the bibliographic coupling between two documents.

**The CSE and the GSE.** The following steps were performed to conduct a comparative analysis between the CSE and the GSE:

- We sourced the complete Scopus bibliographic profile of each Nobel Prize laureate during the same period as the CSE: 1990–2020, in the science categories (i.e., physics, chemistry, and physiology and medicine), and the *Sveriges Riksbank Prize in Economic Sciences in Memory of Alfred Nobel*. A total of 248 laureate profiles were sourced.

- We re-classified CSE and GSE researchers according to their core discipline/subject categories since awardees in respect of both prizes have a wide range of (under)graduate backgrounds. Moreover, prize categories themselves cover a wide range of disciplines/research areas. For example, Juan Camilo Cárdenas, who belongs to the CSE, graduated from industrial engineering but was awarded his prize in the AAEP-EnvSci category (2009). Another example is Daniel Kahneman, a psychologist who received the Nobel Prize in economics (2002). The re-classification was conducted as follows:

  ○ Each CSE and GSE publication belongs to a single journal. Journals and serial titles are classified using the All Science Journal Classification (ASJC), which is based on the aims and scopes of the title and its content [96].

  ○ We cross-checked the printed ISSN of each journal in which the CSE and GSE researchers have published and matched its ASJC subject area. This cross-check procedure was conducted for each CSE and GSE researcher.

  ○ If a given journal belonged to more than one ASJC subject area, it was randomly assigned.

  ○ We then computed each researcher's publication frequency according to the ASJC subject areas.

  ○ Each researcher was then assigned to one ASJC subject according to the researcher's own most frequent ASJC subject based on his/her publication record. For example, Cárdenas, the abovementioned engineer who was awarded a prize in the EnvSci category, was assigned to the economics, econometrics and finance ASJC subject area since most of his articles were published in journals under that classification.

  ○ Once researchers were re-classified into ASJC subjects, a first filter was applied to ensure that the same ASJC category profiles of the GSE coincided with those of the CSE. We then implemented non-proportional stratified sampling since there were not enough values for each strata (i.e., no balanced classes in the GSE-ASJC profiles) due to the particular disciplinary foci of the GSE. Table 2 shows the bibliometric descriptives of the selected laureates. The top three disciplines in the CSE were: agricultural and biological sciences;

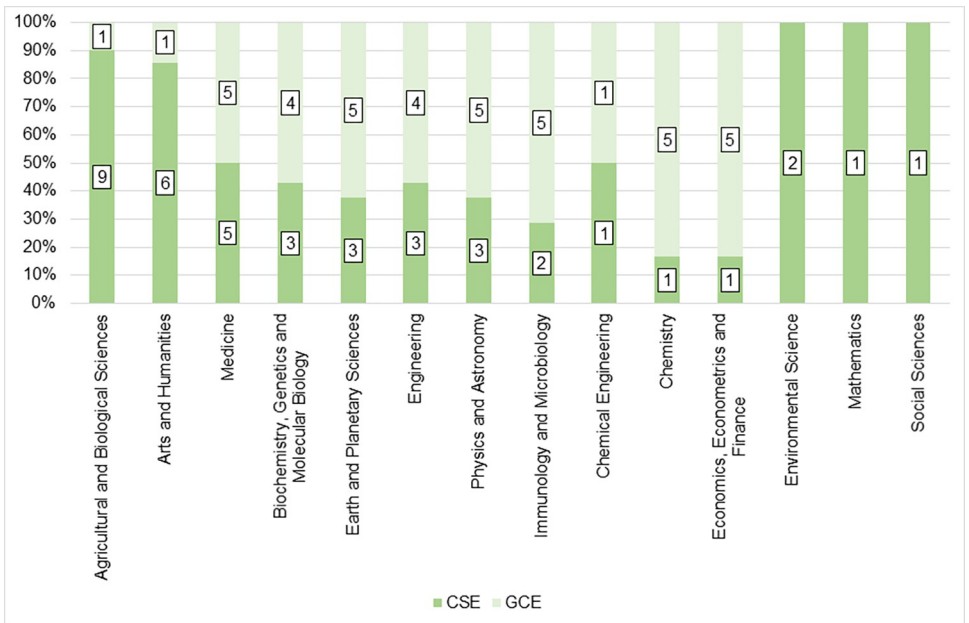

**Fig 1. ASJC subject area classifications and supergroup for CSE and GSE.** Source: the authors based on [78, 96].

arts and humanities; and medicine. In the GSE, each of the following disciplines/fields had five researchers: medicine; earth and planetary sciences; physics and astronomy; immunology and microbiology; chemistry; and economics, econometrics and finance. Fig 1 displays the ASJC subject area classification and grouping for CSE and GSE.

- Since our sample is composed of researchers from multiple disciplines, we had to keep in mind the disciplinary differences so as not to commit methodological flaws, such as comparing the output and citations traditions and research dynamics between two researchers from physics and anthropology. For example, almost ¾ of research references in the humanities refer to books, whereas 80% of those in the natural sciences refer to journal articles [97]. Accordingly, we replicated multiple citation indicators and their composite proposed by Ioannidis et al. [55] to compare individual researchers across different fields. In their seminal study, Ioannidis et al. [55] computed the composite citation indicator (henceforth: $Ci$) for 84,000+ scientists from 12 fields (physics, mathematics, computer science, chemistry, earth sciences, engineering, biology/biotechnology, infectious disease, medicine, brain research, health sciences, and social sciences). $Ci$ has also been used to compare software engineering with the multiple sub-fields of 'information & communication technologies;' to assess the scholarship of media experts on COVID-19 by gender and country; and to estimate the publishing output of COVID-19 and infectious disease experts across 174 research sub-fields [98–100]. In that line, $Ci$ considers total impact; normalized coauthorship; and author order:

  ○ In the first are the number of citations and h index [101]. The $h$ index is defined as follows: for a set of articles $N$ of an author and defining $c_i$ as the number of citations corresponding to an article $i$ then ordering the set of articles in decreasing order according to the number of citations, formally:

$$h \ index = \max\{i \in N : c_i \geq i\} \tag{5}$$

Source: Hirsch [101].

○ In the second, the *hm* index (i.e., an h index adjustment for coauthored papers) [58]. For a set of articles *N* with $c_i$ the number of citations for the article *i* and $a_i$ the number of corresponding authors, the cumulative sum of the inverse of the number of authors is proposed as the effective rank $r_{eff} = \sum^i \frac{1}{a_i}$. Then, sorting the set of articles in decreasing order according to the number of citations, the *hm* index can be defined as:

$$hm \ index = \max\{r_{eff} \in N : c_i \geq r_{eff}\} \qquad (6)$$

Source: [58, 102].

○ In the third, the number of citations as a single author; as a single or first author; and as a single, first or last author.

- Finally, *Ci* was calculated as the sum of the 0–1 normalization log-transformation of the previous indices. Authorship order is crucial when assigning credit/contribution to a research publication. With the exception of mathematics or economics [103], the most credit assigned to a multi-authored article goes to the first author (i.e. early-career researcher), and last author (usually a mentorship figure) [104]. Middle authors generally play a more specific/technical role (i.e., statistical analysis). In sum, *C* brings a more nuanced perspective of an author's impact by including total impact, normalized coauthorship, and the author order as a proxy of the leading role (or absence of it). Table 2 shows Scopus author profiles and bibliometric descriptive of the GSE aggregated and by category 1996–2020.

## Results

### AAEP research topics

Fig 2 displays the semantic networks of the titles of the 88 CSE by category. In SoSci, high betweenness key terms were those related to history (*century*) and development. The two most populated clusters, grouping ~27% of key terms, were related to indigenous peoples/territories; and historical and territorial perspectives on political and social movements. In EnvSci, high betweenness terms were those related to territories and ecology. The two most populated clusters, grouping ~39% of key terms, were related to local natural reserves management strategies; and tools for conservation and identification of biodiversity in the Amazon. In PhySci, high betweenness key terms were those related to control and Alzheimer's. The two most populated clusters, grouping ~31% of key terms, were related to genetics, and research on Alzheimer's and Parkinson's. In all categories, the key term with the highest betweenness was Colombia, highlighting the importance of research for local problems/understanding regardless of category.

**Output and citations.** Fig 3 displays the total articles and citations per year by category. In the three different areas awarded by the AAEP, production was led by the PhySci category, followed by EnvSci and SoSci. This is expected given the output and citation dynamics differential between PhySci and SoSci on inclusion and participation in international journal indexing systems. In the case of citations, there is a well-defined peak produced by the impact of researchers such as Nubia Muñoz, whose primary research is on the human papillomavirus (HPV). In 2003, Muñoz published over 50 articles with ~14,000 citations, turning her into one of the most productive and impactful researchers among the CSE. Fig 3 shows the PhySci category citations with and without (dotted line) the inclusion of Nubia Muñoz. Table 3 presents the most-cited article by AAEP category. All articles were published in internationally

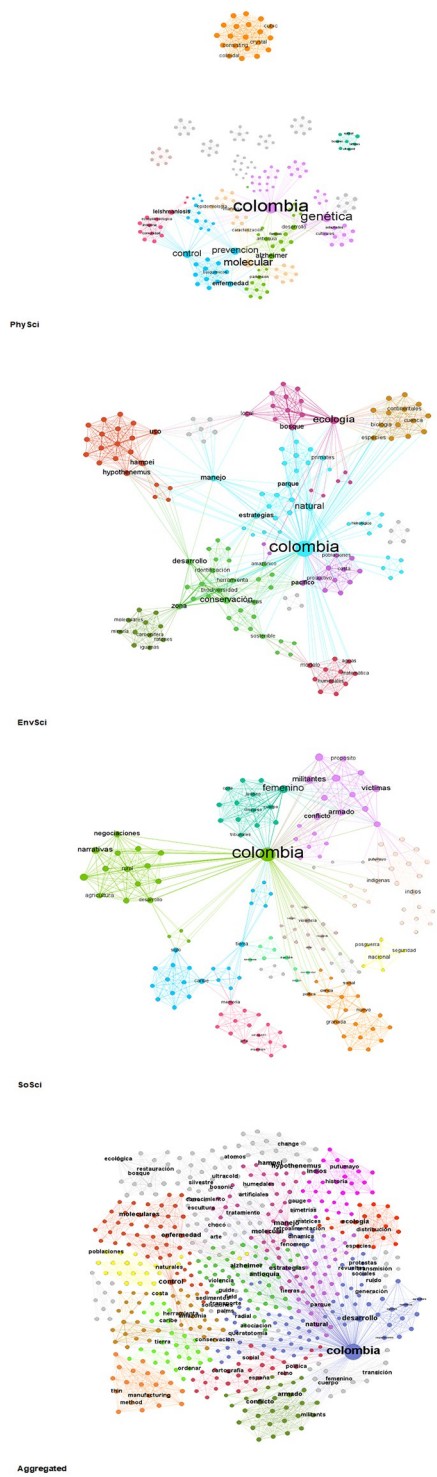

**Fig 2. Semantic networks of research titles awarded by the AAENF by category.** Source: the authors based on AAEF [48, 53]. Processed with quanteda, igraph and gephi [84, 105–107].

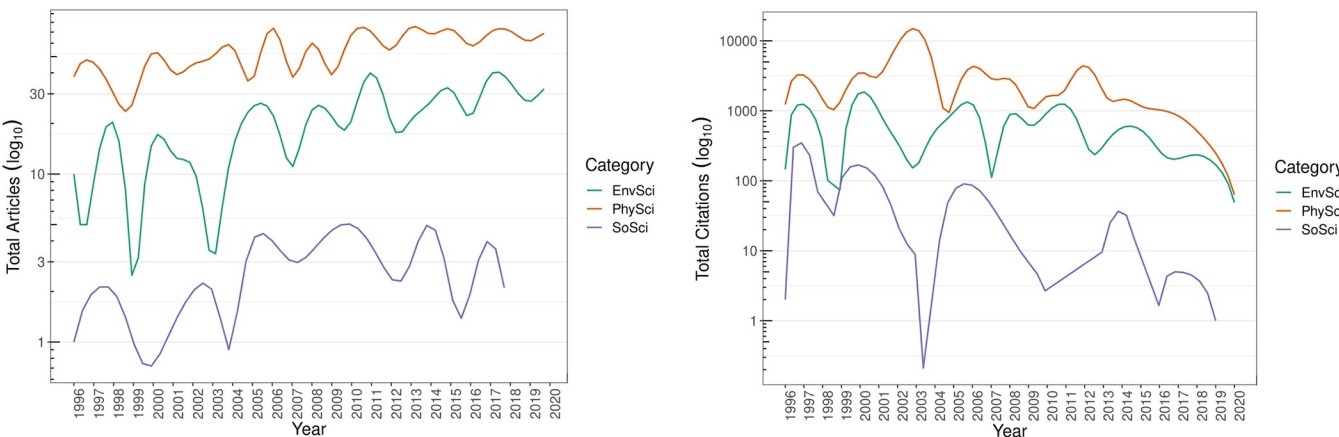

**Fig 3. Total articles (left side) and citations (right side) per year by category.** Source: the authors based on AAEF [48, 53] and Scopus [108]. Processed with quanteda, igraph and gephi [84, 105–107].

reputable journals, edited by either world-renowned universities (Duke University) or societies (*Massachusetts Medical Society*).

Fig 4 shows the citation per article by category of the top three most-cited researchers. Dotted lines indicate the year in which each researcher received the AAEP. First, in the discipline of PhySci, the top three researchers were: Nubia Muñoz (AAEP-2006); Ana María Rey (AAEP-2007); and Iván Darío Vélez (AAEP-2003). As previously stated, Muñoz was the most prolific author in this category, with a peak of citations per paper in 2003 and another peak in 2007, just after receiving the AAEP. However, this metric shows a decrease with no crucial peaks thereafter. Rey shows three peaks after receiving the award. Vélez received the award comparatively early given that his most crucial peak occurred nearly nine years later (2012). Thus, Muñoz appears to have been awarded at the peak of her career, whereas Rey and Vélez were both awarded before their most impactful years.

Second, in EnvSci, the top three researchers were: Germán Poveda (AAEP-2007); Juan Camilo Cárdenas (AAEP-2009); and Consuelo Montes (AAEP-2002). Poveda was awarded after his third career peak. Two additional peaks—although lower—occurred later in his career. Cárdenas followed a similar trend. He was awarded after three career peaks, 2000 being the most significant, followed by a smaller peak in 2012. Montes was awarded in the middle of her first peak, followed by two similar peaks (2006 and 2009). Thus, whereas Poveda and Cárdenas received their awards after having reached their most important peaks, Montes had several post-award peaks. The SoSci category does not allow for much discussion. Suffice to say that after receiving the AAEP, Londoño, and Castillejo-Cuéllar appear to have increased their intermittent involvement in publishing Scopus-indexed articles.

**Table 3. Most-cited article by AAEP category.**

| Category | Author | Article | Journal | Journal H index | Citations |
|---|---|---|---|---|---|
| **PhySci** | Nubia Muñoz | Epidemiologic Classification of Human Papillomavirus Types Associated with Cervical Cancer | *The New England Journal of Medicine* | 1,030 | 4,583 |
| **EnvSci** | Jesús Olivero-Verbel | Repellent activity of essential oils: A review | *Bioresource Technology* | 294 | 631 |
| **SoSci** | Alejandro Castillejo-Cuéllar | Knowledge, Experience, and South Africa's Scenarios of Forgiveness | *Radical History Review* | 22 | 22 |

Source: the authors based on Scopus [108].

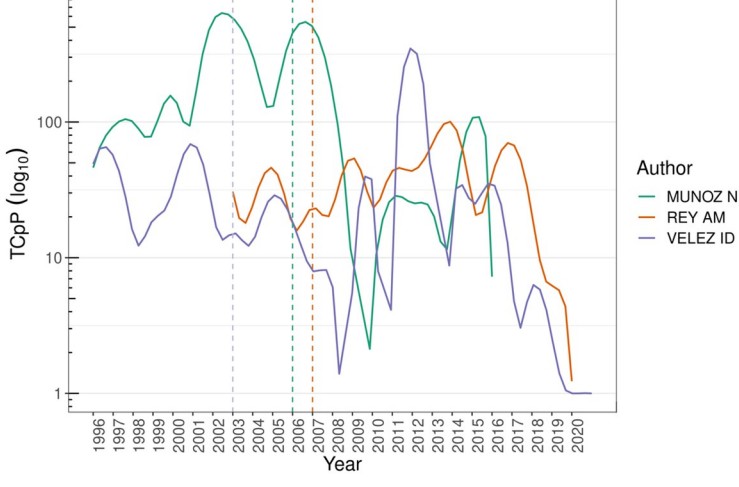

**PhySci**

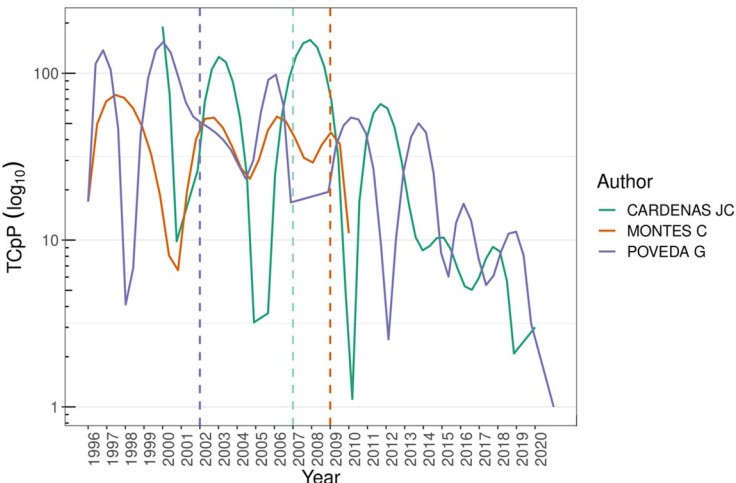

**EnvSci**

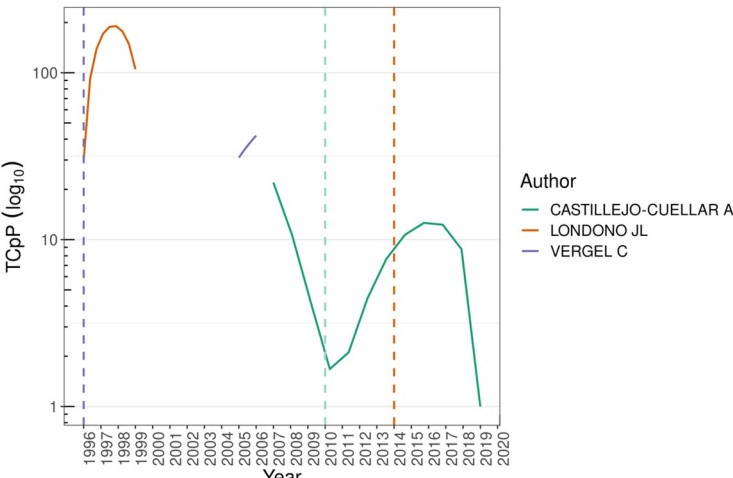

**SoSci**

**Fig 4. Citations per article by category of top three most-cited authors.** Source: the authors based on AAEF [48, 53] and Scopus [108]. Processed with quanteda, igraph and gephi [84, 105–107]. Note: the dashed line indicates the year each author was awarded the AAEP; TCpP: citations per paper.

**Institutional collaboration and coauthorship analysis.** The CSE-PhySci institutional collaboration network has a density of 0.017 (Fig 5). The average path length shows that two random institutions need ~3 steps between middle institutions to reach each other via the shortest path. The network's principal component comprises 26.98% nodes, followed by the remaining clusters with 26.84%, 23.37%, and 12.80%, respectively. Five important Colombian universities ranked among the top-ten institutions with highest betweenness: three public, two private. The institution with the highest betweenness was Universidad de Antioquia (Colombia–Public). Researchers such as Iván Darío Vélez (PhySci-1994, 2003) are currently affiliated with this institution. In contrast, the remaining institutions are international universities or institutions such as the International Agency for research on Cancer (France) or the Institut Catal D'oncologia (Spain), where Nubia Muñoz conducts her research on the human papillomavirus. World-renowned universities such as Harvard University (USA–Private), and Free University of Berlin (Germany–Public) also ranked among the top-ten.

The CSE-EnvSci institutional collaboration network (Fig 6) has a density of 0.031, higher than PhySci, but a lower average path length of 2.6. The principal component is composed of 83.8% of nodes, followed by clusters with 7.3 and 2.2%. The institution with the highest betweenness is the National University of Colombia (Public), one of the most prestigious universities in the country. Some authors like Germán Poveda-EnvSci are affiliated with this university. Among the list are seven major Colombian universities, while the remaining institutions are either international universities or institutions. Among the latter can be found

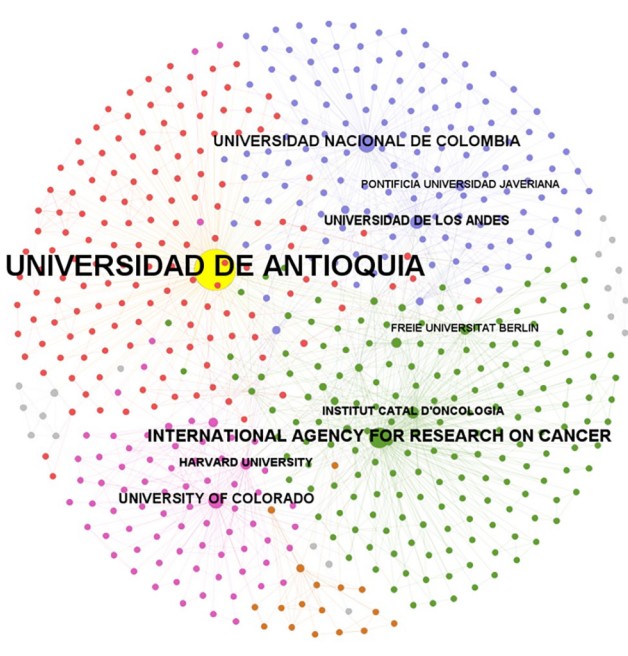

| Cluster color | % nodes |
|---|---|
| | 26.98% |
| | 26.84% |
| | 23.37% |
| | 12.80% |
| | 2.36% |

| Macro indices | |
|---|---|
| Density | 0.01 |
| Average Path Length | 2.94 |

| Institution | Betweenness |
|---|---|
| Universidad de Antioquia | 0.50 |
| International Agency for Research on Cancer | 0.22 |
| Universidad Nacional de Colombia | 0.16 |
| University of Colorado | 0.12 |
| Universidad de Los Andes | 0.11 |
| Institut Catal D'oncologia | 0.09 |
| Harvard University | 0.08 |
| Pontificia Universidad Javeriana | 0.06 |
| Freie Universität Berlin | 0.05 |
| Universidad Del Valle | 0.05 |

**Fig 5. Institutional collaboration network–PhySci.** Sources: the authors based on [78, 105].

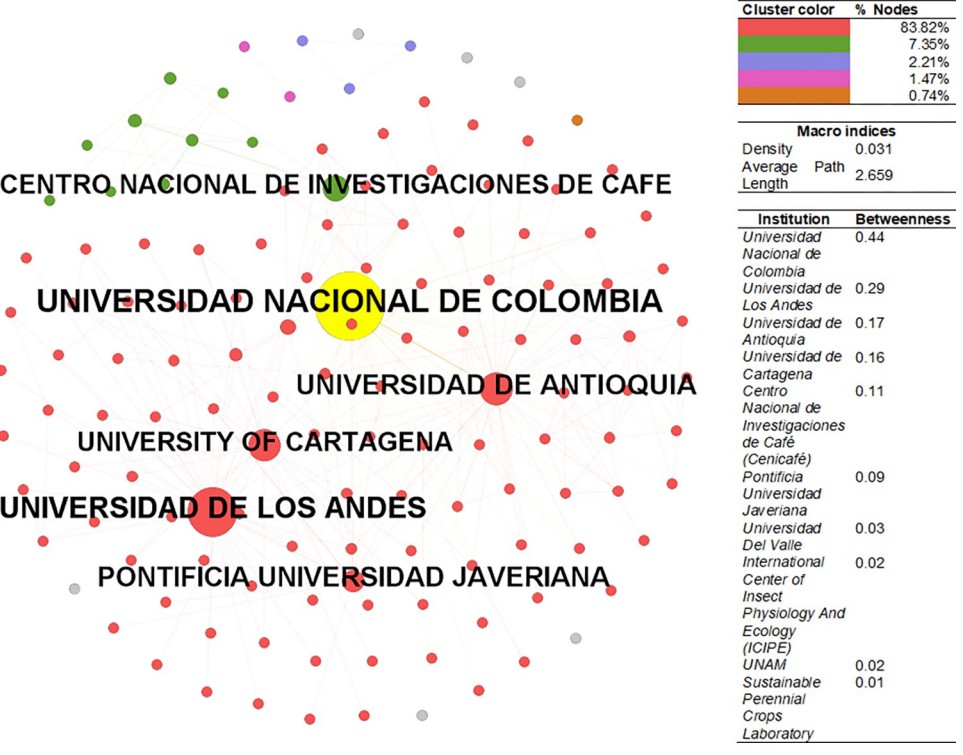

**Fig 6. Institutional collaboration network–EnvSci.** Sources: the authors based on [78, 105].

ICIPE (Kenya), Universidad Nacional Autónoma de México (Public), and the Sustainable Perennial Crops Laboratory (USA), which belongs to the Department of Agriculture.

The CSE-SoSci institutional collaboration network (Fig 7) has a density of 0.133, higher than those in PhySci and EnvSci, but composed of just ten nodes. The average path length is 1.25. The principal component is composed of 40% of nodes, followed by a cluster with 20% of the nodes and two with 10%. The institution with the highest betweenness is Universidad de Los Andes (Colombia–Private), among Colombia's most prestigious private universities. Authors such as Carl Henrik Langebaek (SoSci-2009) is affiliated with it. The remaining universities do not have betweenness properties. There is only one international university: the University of California (USA). Despite the CSE-institutional collaboration being composed mainly of local institutions, they also have a high betweenness of reputable international institutions—in some cases, among the world's top-tier—particularly in the PhySci and EnvSci categories.

The CSE- PhySci coauthorship network (Fig 8) has a density of 0.007. The average path length of 3.62 means that ~4 steps on average are needed for two random nodes to reach each other via the shortest path. The network's principal component comprises 31.95% of the nodes, followed by the remaining clusters with 16.81%, 14.57%, 8.49%, and 7.45%, respectively. Muñoz has the highest betweenness within the principal component. In the second place is Luis Fernando García (AAEP-2000), a physician affiliated with Universidad de Antioquia. Mauricio Restrepo, within Muñoz's cluster, serves as a bridge between the Muñoz and García clusters. Restrepo, on his part, is affiliated with the National Institute of Health, Colombia, and connects the Muñoz and Felipe Guhl clusters. Guhl (AAEP-1998) is a biologist at Universidad de Los Andes. Finally, the Ana María Rey cluster (AAEP-2007) is a closed network. Rey

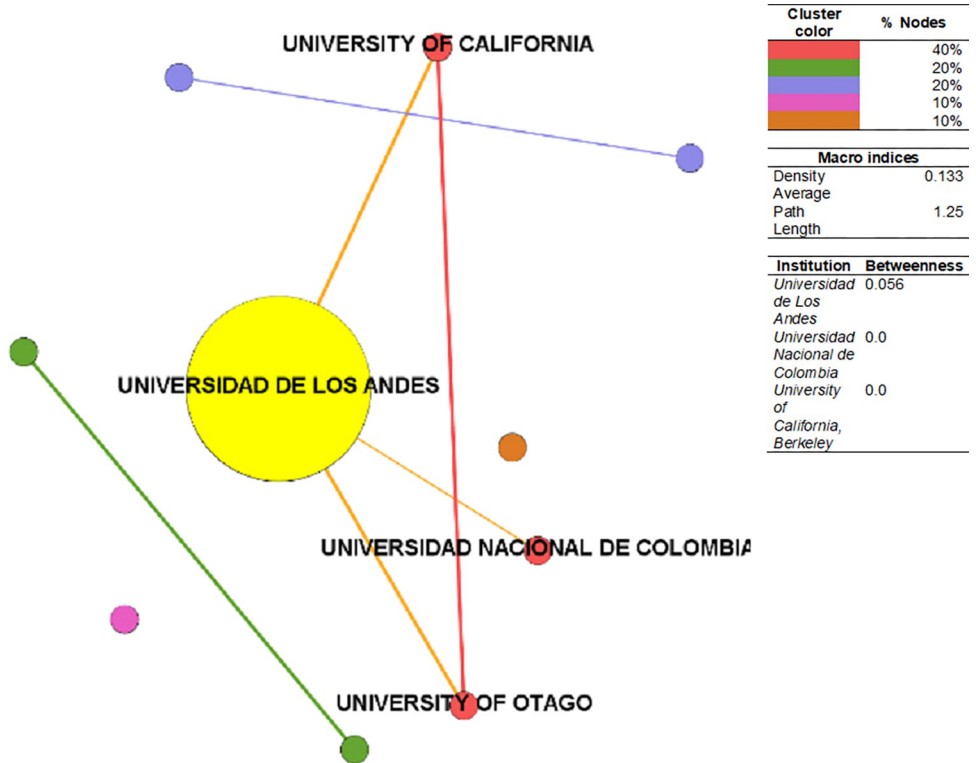

**Fig 7. Institutional collaboration network–SoSci.** Sources: the authors based on [78, 105].

currently works on quantum physics and ultra-cold atoms at the University of Colorado Boulder, a field distant from (micro)biology, genetics, and other branches of medicine.

The CSE-EnvSci network (Fig 9) has a density of 0.019. The average path length is higher at 4.25. The network's principal component comprises 42.82% of the nodes, followed by the remaining clusters comprising 18.48%, 17.89%, 6.74%, and 4.99%, respectively. The node with the highest betweenness is Germán Poveda (AAEP- 1999, 2007, 2019), affiliated with Universidad Nacional de Colombia, Medellín. He specializes in hydraulics. Óscar José Mesa (AAEP-2000, 2007), with the same affiliation as Poveda, ranked in 2nd place. He also works in hydraulics. Authors such as Jaime Carmona-Fonseca (epidemiologist-virologist) or Walter Salas-Zapata (bacteriologist), both affiliated with Universidad de Antioquia, have a higher betweenness despite being outside the five main clusters. In sum, most of the authors are affiliated with local universities, compared to the CSE- PhySci.

The CSE-SoSci network (Fig 10) has a density of 0.128 and an average path length of 1.043, which is the shortest in the study sample. It is also the smallest network, within which the flow of information is the most efficient. The principal component of the network comprises 33.33% of the nodes, followed by the remaining clusters with 14.81%, 11.11%, and 7.41%, respectively. The author with the highest betweenness is Carl Henrik Langebaek (AAEP-2009), an anthropologist affiliated with Universidad de Los Andes. In the same cluster can be found Melanie J. Miller, an anthropologist at University of Otago, New Zealand; and Sabrina C. Agarwal, an anthropologist at University of California, Berkeley. Both work in similar bioarchaeological fields.

**Bibliographic coupling networks.** The bibliographic coupling network clusters were labeled according to the most frequent ASJC subject among each journal cluster. If the most

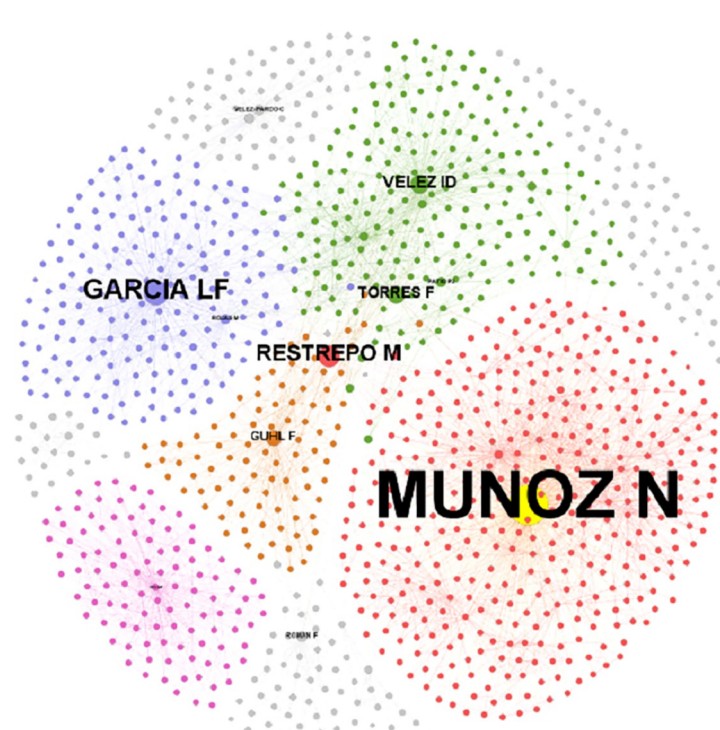

**Fig 8. Author collaboration network–PhySci.** Sources: the authors based on [78, 105].

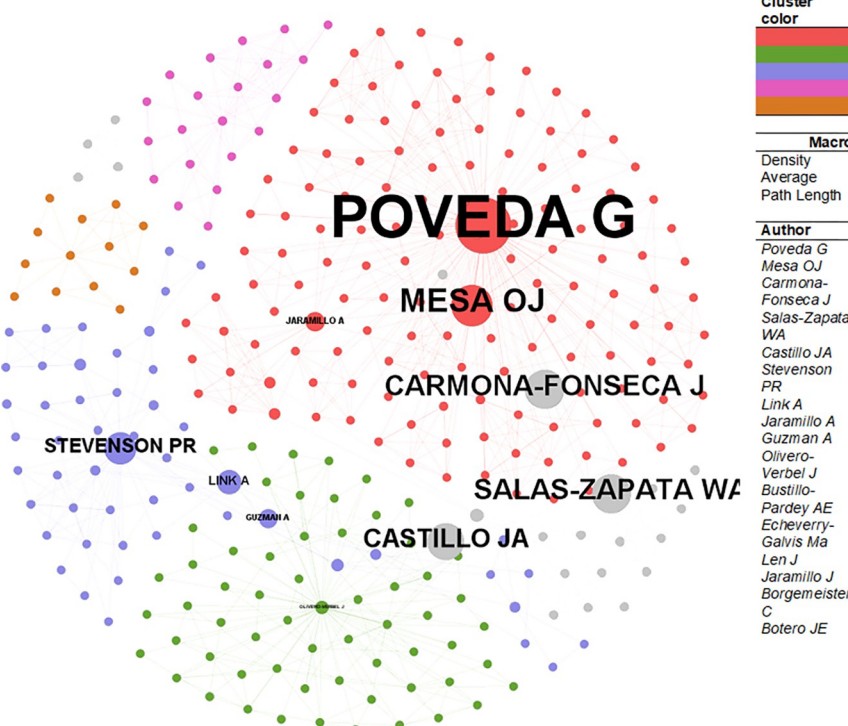

**Fig 9. Author collaboration network–EnvSci.** Sources: the authors based on [78, 105].

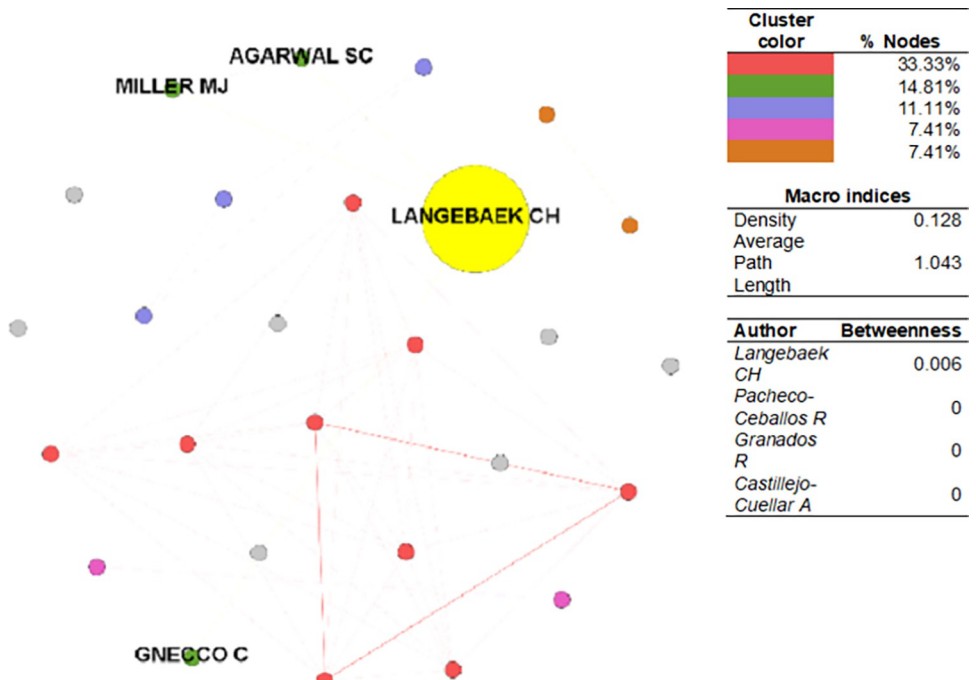

**Fig 10. Author collaboration network–SoSci.** Sources: the authors based on [78, 105].

frequent ASJC subject in a given cluster reaches at least half of the first most frequent subject, that cluster will also share the label of the second most frequent subject. Nodes with a degree less than five were hidden from the layout to improve the structure's interpretation.

Fig 11 presents the PhySci bibliographic coupling network. Density equals 0.026 and the average path length is 3.56, which means an average of ~4 steps for a random pair of nodes to reach each other via the shortest path, those nodes being a pair of coupled documents. The principal component of this network corresponds to: medicine/agricultural and biological sciences having 43.53% of the network's nodes, followed by physics and astronomy with 20.98%; biochemistry, genetics and molecular biology with 12.50%; 6.11% for immunology and microbiology; and 2.95% for mathematics. Even with the principal component mainly comprising articles classified in medicine, most articles with the highest property of mediating the flow/share of knowledge/references belonged to physics—with a marginal presence of articles on movement disorders (e.g., Parkinson's disease).

Fig 12 presents the EnvSci bibliographic coupling network. The density equals 0.058 with an average path length of 2.56. Most of the network corresponds to the agricultural and biological sciences/earth and planetary sciences with 85.77% nodes, followed by 1.80% for chemical engineering. Documents with the highest betweenness were mostly articles in (sub)fields such as contamination and toxicology, ecology, geology, environmental economics, and molecular catalysis. There is also the involvement of a report from the IPCC-2014 (Intergovernmental Panel on Climate Change).

Fig 13 presents the SoSci bibliographic coupling network. The network has a density of 0.051 with an average path length of 2.339. The principal component is composed mainly of arts and humanities/social sciences, with 45.28% of nodes. The document with the highest betweenness was on the history of land-use planning in Colombia. Most journals were devoted

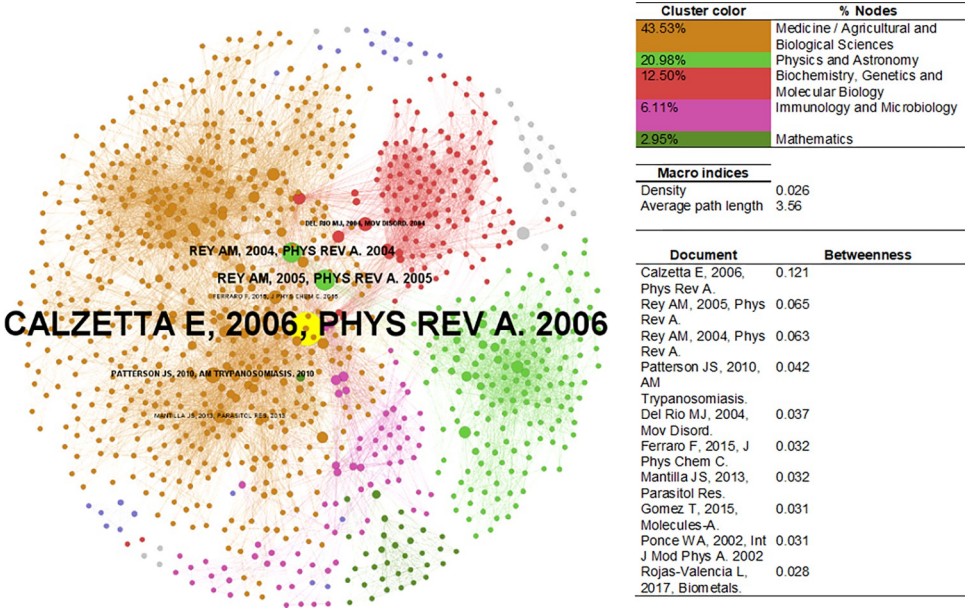

| Cluster color | | % Nodes | |
|---|---|---|---|
| | 43.53% | Medicine / Agricultural and Biological Sciences | |
| | 20.98% | Physics and Astronomy | |
| | 12.50% | Biochemistry, Genetics and Molecular Biology | |
| | 6.11% | Immunology and Microbiology | |
| | 2.95% | Mathematics | |

| Macro indices | |
|---|---|
| Density | 0.026 |
| Average path length | 3.56 |

| Document | Betweenness |
|---|---|
| Calzetta E, 2006, Phys Rev A. | 0.121 |
| Rey AM, 2005, Phys Rev A. | 0.065 |
| Rey AM, 2004, Phys Rev A. | 0.063 |
| Patterson JS, 2010, AM Trypanosomiasis. | 0.042 |
| Del Rio MJ, 2004, Mov Disord. | 0.037 |
| Ferraro F, 2015, J Phys Chem C. | 0.032 |
| Mantilla JS, 2013, Parasitol Res. | 0.032 |
| Gomez T, 2015, Molecules-A. | 0.031 |
| Ponce WA, 2002, Int J Mod Phys A. 2002 | 0.031 |
| Rojas-Valencia L, 2017, Biometals. | 0.028 |

**Fig 11. Bibliographic coupling network–PhySci.** Sources: the author based on [78, 105].

to history and anthropology, and geography, with a few exceptions in law, health policy, and hydraulic engineering.

Fig 14 presents a summary of the nodes and macro indices of the three types of networks presented above. The number of nodes is proportional to the number of articles in each CSE category. In contrast, the density is inversely proportional to the number of articles. The decreasing average path length reinforces this observation.

In terms of institutional collaboration, the SoSci network has established more real than potential institutional collaborations, followed by EnvSci and PhySci. Accordingly, there are fewer intermediates between a pair of institutions than PhySci and EnvSci. On the other hand, the number of institutions in PhySci is almost ~6 times higher than EnvSci, and ~49 times higher than SoSci, giving PhySci a higher average path length. Compared to EnvSci, however, PhySci does not have over a complete intermediate institution in average. In contrast, there are on average ~2 different intermediate institutions between PhySci and SoSci. The coauthorship networks display similar patterns regarding the average number of intermediate authors for PhySci and SoSci. However, the EnvSci network showed the highest average path length with ~4 middle authors.

Among the top-ten institutions, the PhySci collaboration network showed a direct collaboration between Universidad de Antioquia and Harvard University (161 Nobel Prizes). In EnvSci, Universidad de Los Andes, Nacional, Cartagena, and Valle, have at least one direct collaboration with UNAM (3 Nobel Prizes). In SoSci, Universidad de Los Andes has a direct collaboration with the University of California, Berkeley (110 Nobel Prizes). Thus, CSE institutions are embedded in a collaboration network with GSE institutions. In a more refined analysis at the authorship level, only two AAEPs have coauthored with GSE authors: Juan Camilo Cárdenas with Elinor Ostrom in *What do people bring into the game? Experiments in the field about cooperation in the commons* (2004); and Nubia Muñoz with Harald zur Hausen in *Nasopharyngeal carcinoma. X. Presence of epstein-barr genomes in separated epithelial cells of tumours in patients from Singapore, Tunisia and Kenya* (1975).

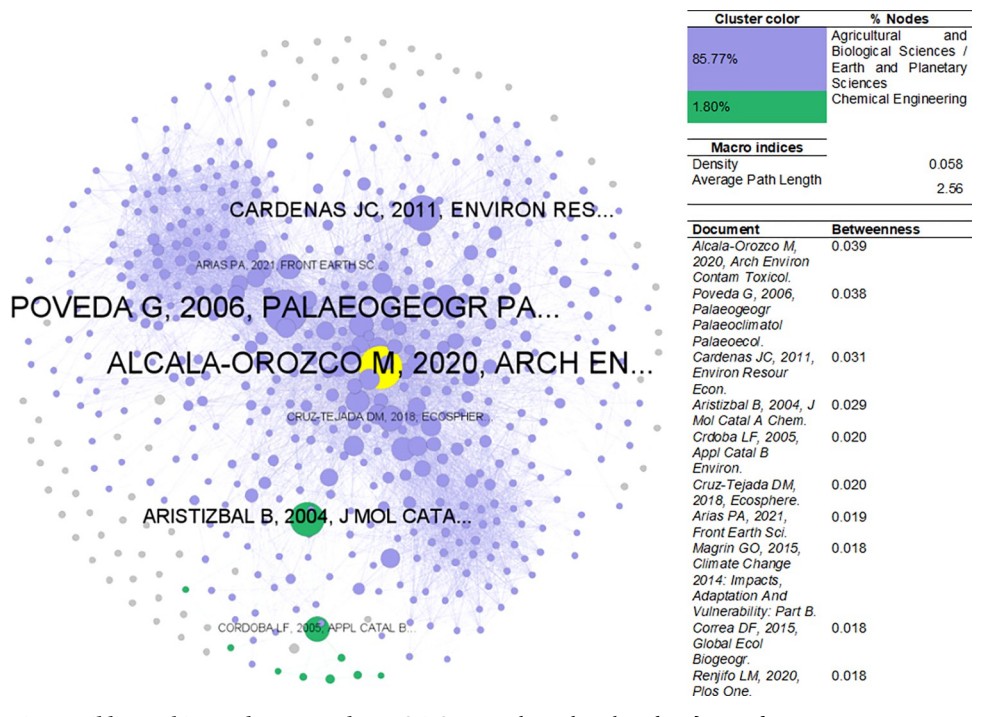

**Fig 12. Bibliographic coupling network–EnvSci.** Sources: the authors based on [78, 105].

Regarding bibliographic coupling networks, the PhySci network displayed a more diverse topic-cluster formation in terms of research fronts (i.e., shared knowledge/references) than the more multi/inter/transdisciplinary categories of EnvSci. The SoSci network was the most homogeneous network. In SoSci, despite the reduced number of research fronts, the average

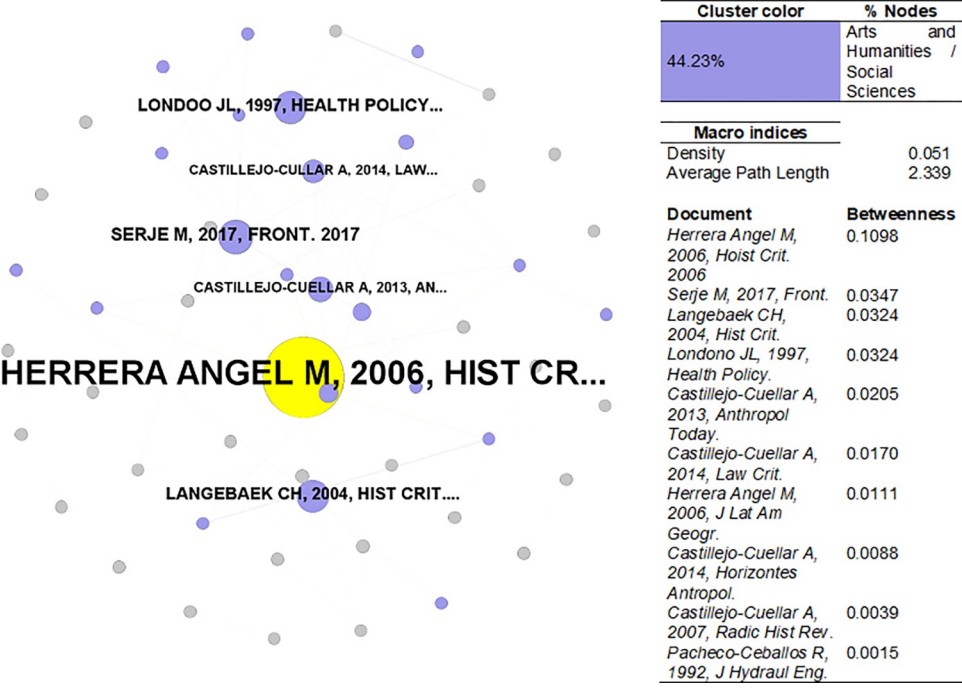

**Fig 13. Bibliographic coupling network–SoSci.** Sources: the authors based on [78, 105].

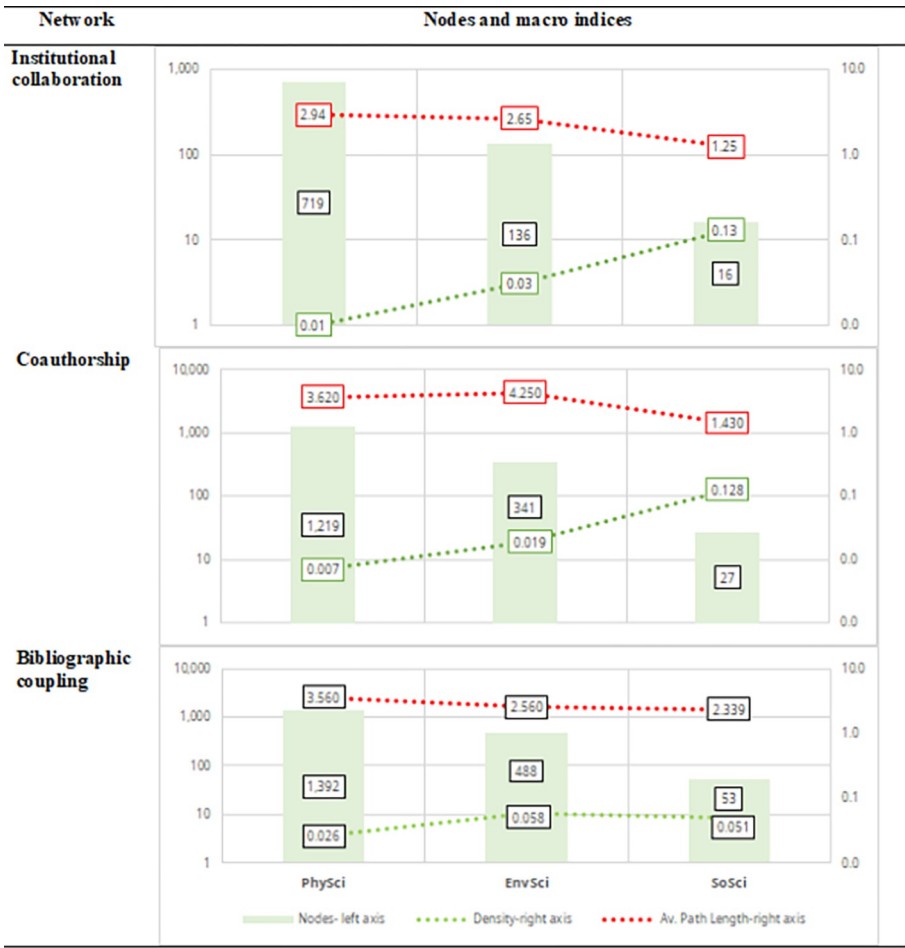

**Fig 14. Nodes and macro indices summary.** Sources: the authors based on [78, 105].

middle document was ~2, similar to EnvSci, although EnvSci has ~9 times the number of nodes.

**The CSE and the GSE.** Table 4 presents the citation indicators and $Ci$ [55]. GSE was yellow while CSE yellow-light colored. The six scientific impact and productivity indices are separated into two aspects: bulk impact (total number of citations [NC] and h index [H]); and authorship order adjusted impact (Schreiber Hm index [HM]; total citations for papers where the scientist is the single author [NS]; total citations for papers where the scientist is the single or first author [NSF]; and total citations for papers where the scientist is the single, first, or last author [NSFL]). As a reminder, $Ci$ is calculated as the sum of the normalized log-transformation of previous indexes. The normalization of 0–1 range values was computed according to each CSE and GSE. Each column is colored from dark (higher score) to light green (lower).

After dividing $Ci$ into quartiles, Nubia Muñoz (medicine) is the lone AAEP among the 4th quartile. She also ranked 1st in the NSF indicator. The appearance of Muñoz in all the sections presented above, even in a direct comparison with the GSE, is explained by her contribution to the study of the human papillomavirus, which earned her a Nobel Prize nomination by the International Epidemiological Association. The highest was that of Alan J. Heeger (chemistry). Among the 3rd quartile, several AAEPs emerged: Germán Poveda Jaramillo (earth and

**Table 4. Citation indicators and *Ci* for the CSE and GSE.**

| Rank | Award | Discipline/Field | Author | NC | H | HM | NS | NSF | NSFL | *Ci* |
|------|-------|------------------|--------|-----|---|-----|-----|-----|------|------|
| 1 | Nobel | Engineering | Alan J. Heeger | 81,102 | 135 | 58.07 | 5,106 | 5,164 | 59,254 | 5.86 |
| 2 | Nobel | Chemistry | John B. Goodenough | 61,250 | 109 | 60.3 | 3,313 | 15,458 | 49,009 | 5.84 |
| 3 | Nobel | Engineering | Shuji Nakamura | 50,231 | 103 | 47.59 | 3,421 | 15,089 | 38,720 | 5.74 |
| 4 | Nobel | Biochemistry, Genetics and Molecular Biology | Robert H. Grubbs | 61,332 | 115 | 65.37 | 2,206 | 4,720 | 50,214 | 5.7 |
| 5 | Nobel | Chemical engineering | Ben Feringa | 51,112 | 114 | 60.01 | 1,828 | 3,627 | 39,923 | 5.59 |
| 6 | Nobel | Biochemistry, Genetics and Molecular Biology | Aarón Ciechanover | 32,557 | 66 | 35.04 | 2,833 | 5,579 | 26,248 | 5.37 |
| 7 | AAEP | Medicine | Nubia Muñoz | 42,754 | 84 | 27.89 | 1,013 | 15,967 | 20,622 | 5.36 |
| 8 | Nobel | Earth and Planetary Sciences | Paul J. Crutzen | 22,413 | 64 | 29.16 | 3,201 | 4,592 | 11,975 | 5.21 |
| 9 | Nobel | Economics, Econometrics and Finance | Esther Duflo | 19,944 | 62 | 31.43 | 1,920 | 6,131 | 11,400 | 5.18 |
| 10 | Nobel | Earth and Planetary Sciences | James Peebles | 14,077 | 42 | 28.25 | 2,488 | 9,151 | 13,821 | 5.13 |
| 11 | Nobel | Immunology and Microbiology | Jules A. Hoffmann | 24,312 | 72 | 22.23 | 1,060 | 4,210 | 13,069 | 5.05 |
| 12 | Nobel | Immunology and Microbiology | Tasuku Honjo | 36,472 | 93 | 34.12 | 209 | 968 | 23,529 | 4.95 |
| 13 | Nobel | Chemistry | Richard Smalley | 46,843 | 83 | 22.45 | 422 | 724 | 27,874 | 4.93 |
| 14 | Nobel | Engineering | Hiroshi Amano | 31,879 | 81 | 35.13 | 122 | 1,505 | 6,810 | 4.79 |
| 15 | Nobel | Biochemistry, Genetics and Molecular Biology | James E. Rothman | 17,809 | 58 | 22.38 | 350 | 1,542 | 13,569 | 4.75 |
| 16 | Nobel | Physics and Astronomy | Anthony James Leggett | 4,678 | 31 | 22.94 | 3,187 | 3,294 | 4,587 | 4.74 |
| 17 | Nobel | Immunology and Microbiology | Peter C. Doherty | 13,469 | 65 | 28.19 | 195 | 1,547 | 8,649 | 4.69 |
| 18 | Nobel | Immunology and Microbiology | James P. Allison | 31,295 | 77 | 29.73 | 174 | 213 | 22,814 | 4.69 |
| 19 | Nobel | Economics, Econometrics and Finance | Michael Kremer | 8,715 | 46 | 22.87 | 648 | 1,818 | 5,915 | 4.66 |
| 20 | Nobel | Chemistry | Rudolph A. Marcus | 5,359 | 37 | 25.18 | 1,000 | 1,013 | 5,169 | 4.56 |
| 21 | Nobel | Physics and Astronomy | William Daniel Phillips | 8,743 | 42 | 13.23 | 974 | 988 | 5,011 | 4.48 |
| 22 | Nobel | Economics, Econometrics and Finance | Lars Peter Hansen | 5,158 | 40 | 20.25 | 202 | 3,620 | 3,692 | 4.44 |
| 23 | Nobel | Physics and Astronomy | Horst Ludwig Störmer | 34,885 | 48 | 11.91 | 178 | 578 | 7,091 | 4.39 |
| 24 | Nobel | Physics and Astronomy | Robert B. Laughlin | 3,195 | 20 | 13.69 | 900 | 1,527 | 1,660 | 4.18 |
| 25 | AAEP | Earth and Planetary Sciences | Germán Poveda Jaramillo | 4,443 | 37 | 16.64 | 35 | 3,066 | 3,521 | 4.14 |
| 26 | Nobel | Medicine | Michael Houghton | 7,147 | 40 | 13.14 | 312 | 591 | 1,336 | 4.14 |
| 27 | Nobel | Engineering | Hideki Shirakawa | 2,224 | 21 | 14.23 | 692 | 718 | 1,583 | 4.05 |
| 28 | AAEP | Agricultural and Biological Sciences | Pablo R. Stevenson | 1,564 | 22 | 15.48 | 341 | 969 | 1,228 | 3.97 |
| 29 | AAEP | Agricultural and Biological Sciences | Felipe Guhl Nannetti | 2,752 | 34 | 14.85 | 80 | 941 | 1,719 | 3.96 |
| 30 | AAEP | Economics, Econometrics and Finance | Juan Camilo Cardenas Campo | 1,753 | 20 | 11.44 | 344 | 1,274 | 1,526 | 3.94 |
| 31 | Nobel | Medicine | Barry Marshall | 1,867 | 24 | 14.65 | 252 | 411 | 1,377 | 3.88 |
| 32 | Nobel | Earth and Planetary Sciences | F. Sherwood Rowland | 4,170 | 36 | 11.51 | 119 | 119 | 2,410 | 3.82 |
| 33 | Nobel | Chemistry | Yves Chauvin | 1,759 | 20 | 7.86 | 449 | 577 | 973 | 3.77 |
| 34 | Nobel | Economics, Econometrics and Finance | Robert Aumann | 909 | 13 | 10.67 | 516 | 903 | 909 | 3.74 |
| 35 | Nobel | Biochemistry, Genetics and Molecular Biology | John Bennett Fenn | 1,345 | 13 | 7.7 | 576 | 669 | 1,290 | 3.72 |
| 36 | Nobel | Immunology and Microbiology | Françoise Barré-Sinoussi | 3,408 | 30 | 12.12 | 61 | 220 | 524 | 3.62 |
| 37 | AAEP | Biochemistry, Genetics and Molecular Biology | Marlene Jimenez Del Rio | 1,977 | 26 | 15.97 | 9 | 752 | 1,275 | 3.59 |
| 38 | AAEP | Physics and Astronomy | Ana María Rey Ayala | 5,199 | 37 | 17.15 | 0 | 614 | 2,583 | 3.55 |
| 39 | Nobel | Earth and Planetary Sciences | George F. Smoot | 3,158 | 25 | 10.97 | 38 | 59 | 935 | 3.41 |
| 40 | AAEP | Biochemistry, Genetics and Molecular Biology | Carlos Alberto Vélez Pardo | 2,191 | 27 | 16.48 | 0 | 546 | 1,892 | 3.36 |
| 41 | AAEP | Environmental science | Jesus Olivero Verbel | 2,773 | 27 | 15.17 | 0 | 683 | 1,389 | 3.35 |
| 42 | AAEP | Medicine | Iván Darío Vélez Bernal | 6,021 | 31 | 12.11 | 0 | 399 | 1,159 | 3.33 |
| 43 | Nobel | Arts and Humanities | Thomas Schelling | 339 | 10 | 9.44 | 261 | 261 | 320 | 3.26 |
| 44 | AAEP | Agricultural and Biological Sciences | Luis Miguel Renjifo Martínez | 461 | 7 | 4.74 | 299 | 306 | 438 | 3.12 |
| 45 | Nobel | Economics, Econometrics and Finance | James Mirrlees | 425 | 6 | 5.37 | 277 | 315 | 315 | 3.08 |
| 46 | AAEP | Mathematics | Federico Ardila Mantilla | 416 | 11 | 7.53 | 33 | 416 | 416 | 3.08 |
| 47 | Nobel | Chemistry | Makoto Kobayashi | 2,395 | 27 | 8.44 | 0 | 308 | 423 | 3.02 |

*(Continued)*

**Table 4.** (Continued)

| Rank | Award | Discipline/Field | Author | NC | H | HM | NS | NSF | NSFL | *Ci* |
|------|-------|-----------------|--------|-----|---|-----|-----|-----|------|------|
| 48 | Nobel | Medicine | Georges Charpak | 1,302 | 11 | 3.7 | 19 | 183 | 1,122 | 2.97 |
| 49 | Nobel | Medicine | Paul Lauterbur | 1,033 | 12 | 6.28 | 35 | 45 | 583 | 2.95 |
| 50 | Nobel | Agricultural and Biological Sciences | William C. Campbell | 230 | 6 | 6.33 | 195 | 201 | 230 | 2.94 |
| 51 | AAEP | Immunology and Microbiology | Luis Fernando Garcia | 4,130 | 37 | 15.1 | 0 | 0 | 2,224 | 2.82 |
| 52 | Nobel | Earth and Planetary Sciences | Riccardo Giacconi | 1,092 | 12 | 4.09 | 30 | 35 | 351 | 2.77 |
| 53 | Nobel | Medicine | Jens C. Skou | 190 | 4 | 4.5 | 175 | 190 | 190 | 2.73 |
| 54 | AAEP | Biochemistry, Genetics and Molecular Biology | Fernando Echeverri Lopez | 914 | 20 | 6.64 | 0 | 118 | 454 | 2.72 |
| 55 | AAEP | Agricultural and Biological Sciences | Juliana Jaramillo Salazar | 598 | 13 | 5.13 | 0 | 533 | 570 | 2.71 |
| 56 | AAEP | Physics and Astronomy | William A. Ponce Gutiérrez | 623 | 13 | 8.28 | 0 | 257 | 374 | 2.71 |
| 57 | AAEP | Social sciences | Alejandro Castillejo Cuéllar | 92 | 6 | 7 | 92 | 92 | 92 | 2.63 |
| 58 | AAEP | Chemical engineering | Consuelo Montes De Correa | 852 | 17 | 6.81 | 0 | 66 | 423 | 2.62 |
| 59 | AAEP | Chemistry | Jhon Fredy Perez Torres | 360 | 12 | 5.58 | 28 | 54 | 59 | 2.61 |
| 60 | Nobel | Physics and Astronomy | François Englert | 257 | 9 | 4.9 | 22 | 76 | 80 | 2.52 |
| 61 | AAEP | Agricultural and Biological Sciences | Nubia Estela Matta Camacho | 372 | 12 | 4.79 | 0 | 65 | 239 | 2.33 |
| 62 | AAEP | Earth and Planetary Sciences | Oscar José Mesa Sánchez | 1,416 | 13 | 7.05 | 0 | 0 | 1,020 | 2.26 |
| 63 | AAEP | Immunology and Microbiology | Pablo J. Patiño Grajales | 407 | 12 | 3.11 | 0 | 74 | 182 | 2.23 |
| 64 | AAEP | Earth and Planetary Sciences | Andrés Alejandro Plazas Malagón | 137 | 7 | 2.58 | 1 | 101 | 101 | 1.96 |
| 65 | AAEP | Arts and Humanities | Diana Obregón Torres | 33 | 3 | 4 | 33 | 33 | 33 | 1.94 |
| 66 | AAEP | Medicine | Alberto Gómez Gutiérrez | 249 | 9 | 4.1 | 0 | 14 | 57 | 1.91 |
| 67 | AAEP | Engineering | Francisco José Román Campos | 129 | 6 | 4.18 | 0 | 26 | 114 | 1.9 |
| 68 | AAEP | Medicine | Walter Alfredo Salas Zapata | 82 | 6 | 3.08 | 0 | 77 | 77 | 1.86 |
| 69 | AAEP | Physics and Astronomy | Cristian Edwin Susa Quintero | 101 | 7 | 3.12 | 0 | 50 | 63 | 1.85 |
| 70 | AAEP | Arts and Humanities | Astrid Ulloa | 16 | 3 | 3 | 16 | 16 | 16 | 1.58 |
| 71 | AAEP | Medicine | Francisco Lopera Restrepo | 288 | 8 | 3.64 | 0 | 0 | 21 | 1.51 |
| 72 | AAEP | Agricultural and Biological Sciences | Jesús Orlando Vargas Ríos | 83 | 6 | 2.98 | 0 | 0 | 31 | 1.33 |
| 73 | AAEP | Engineering | Juan Carlos Salcedo Reyes | 66 | 6 | 1.64 | 2 | 3 | 3 | 1.15 |
| 74 | AAEP | Agricultural and Biological Sciences | Alex E Bustillo Pardey | 48 | 5 | 2.16 | 0 | 0 | 8 | 1.04 |
| 75 | AAEP | Agricultural and Biological Sciences | Jorge Eduardo Botero | 27 | 3 | 1.25 | 0 | 4 | 21 | 0.99 |
| 76 | AAEP | Arts and Humanities | Mauricio Nieto Olarte | 12 | 2 | 1.92 | 1 | 12 | 12 | 0.99 |
| 77 | AAEP | Arts and Humanities | Sergio Andrés Mejía Macía | 6 | 1 | 2 | 6 | 6 | 6 | 0.87 |
| 78 | AAEP | Engineering | Raul Pacheco Ceballos | 4 | 2 | 2 | 4 | 4 | 4 | 0.84 |
| 79 | AAEP | Environmental science | Margarita Serje De La Ossa | 5 | 1 | 2 | 5 | 5 | 5 | 0.79 |
| 80 | AAEP | Agricultural and Biological Sciences | Carlos Enrique Sarmiento Pinzón | 36 | 2 | 0.81 | 0 | 0 | 0 | 0.41 |
| 81 | AAEP | Arts and Humanities | Carl Henrik Langebaek Rueda | 5 | 1 | 1.5 | 0 | 0 | 4 | 0.35 |
| 82 | AAEP | Arts and Humanities | Marta Herrera Ángel | 2 | 1 | 1 | 2 | 2 | 2 | 0.26 |

Source: the authors based on [55, 78, 96]. Note: NC: total citations; H: H index; Hm: Schreiber Hm index; NS: total citations for papers where the scientist is the single author; NSF: total citations for papers where the scientist is the single or first author; NSFL: total citations for papers where the scientist is the single, first, or last author; *Ci*: composite citation indicator.

planetary sciences); Pablo R. Stevenson (agricultural and biological sciences); Felipe Guhl Nan-netti (agricultural and biological sciences); Juan Camilo Cárdenas (economics, econometrics and finance); Marlene Jiménez Del Rio (biochemistry, genetics and molecular biology); Ana María Rey Ayala (physics and astronomy); and Carlos Alberto Vélez Pardo (biochemistry, genetics and molecular biology). On the other hand, there were several GSEs ranked 2nd and one was ranked 1st, for instance, Thomas Schelling (arts and humanities); James Mirrlees (economics, econometrics and finance); Makoto Kobayashi (chemistry); Georges Charpak

(medicine); François Englert (physics and astronomy). Thus, neither scientific elite was fractured into two mutually exclusive groups. Fig 15 displays the violin-box plot for each index and group.

## Discussion

Most of the CSE has motivations/incentives other than publishing research in academic journals. This is due to the AAEP's broad scope in granting awards. Awards are granted not only for research articles, dissertations (MSc/PhD), and books, but also for technical reports or books published by leading national institutions, such as IDEAM (Institute of Hydrology, Meteorology and Environmental Studies) or the Alexander von Humboldt Biological Resources Research Institute (EnvSci); or the National Centre for Historical Memory; and NGOs such as Tropenbos International (SoSci). In contrast, all GSEs 1990–2020 had publications in Scopus.

Yet, in all science categories, the GSE had a decreasing trend in annual growth output. At the individual level, this explains the temporary dip in GSE impact after winning the award and reduced productivity albeit with a higher impact [11, 26]. The inverse relationship between age and productivity could be another explanatory factor since the average laureate age over the past 25 years is 44.1±9.7, those in physics being the youngest at 42±12.5 [109]. Additional factors, as noted by Diamandis [110] in an *opinion* piece, are as follows: laureates receive the prize 20–50 years after their core contribution to their field, by which time they are past their prime; they become less active in terms of research; or their main contribution was a serendipitous *one-hit-wonder*. Also, we consider deceased Nobel laureates (e.g., Robert H. Grubbs or Paul J. Crutzen). Despite, for obvious reasons, they halt their research output, their influence and impact (citations) in research conducted after their retirement or passing affects their overall composite indicator. At the country level, research on endogenous growth suggests that *ideas are getting harder to find* and that more human and financial resources are required to maintain the same growth levels as those in previous decades (e.g., 18 times the number of researchers are required nowadays to double the chip density than was the case in the 1970s) [111]. Several explanatory factors could be outlined here, such as reduced public funding, *follow on* innovations that produce smaller growth, or the fact that a third industrial revolution led by computers, the Internet, and mobile phones, produced only a stunted growth between 1996–2004 [112].

Conversely, the AAEP showed an increasing trend in all categories. It is important, however, to consider the following growth comparison in the limited context of the study's specific research sample and scale. The annual growth rate for each category ranges between <1% for SoSci and 6.67% for EnvSci. Given that the annual growth rate for modern science post-WWII is ~9%, the overall output growth figure of 2.97% is below such an estimate [113]. The CSE's overall growth was similar to that in the mid-18th century and between WWI and WWII (~3%). In the context of developing countries, the average annual growth 1996–2018 in the fields of PhySci, EnvSci, and SoSci was 18% [49]. Thus, even in the most prolific category (EnvSci), the growth is below historical and disciplinary estimates but positive compared to that of the GSE. Two examples provide details on the CSE output/impact. A special case in terms of output is that of Ana María Rey (PhySci) and Juan Camilo Cárdenas (EnvSci). Rey has coauthored multiple articles in most of the physics journals that publish Nobel Prize research [5] (i.e., *Physical Review; Science;* and *Nature*) and has made contributions to fields such as solid states physics–a field that garnered a significant amount of Nobel Prizes in the 20th century [14]. Cárdenas also has coauthored articles published in *Science*. His research

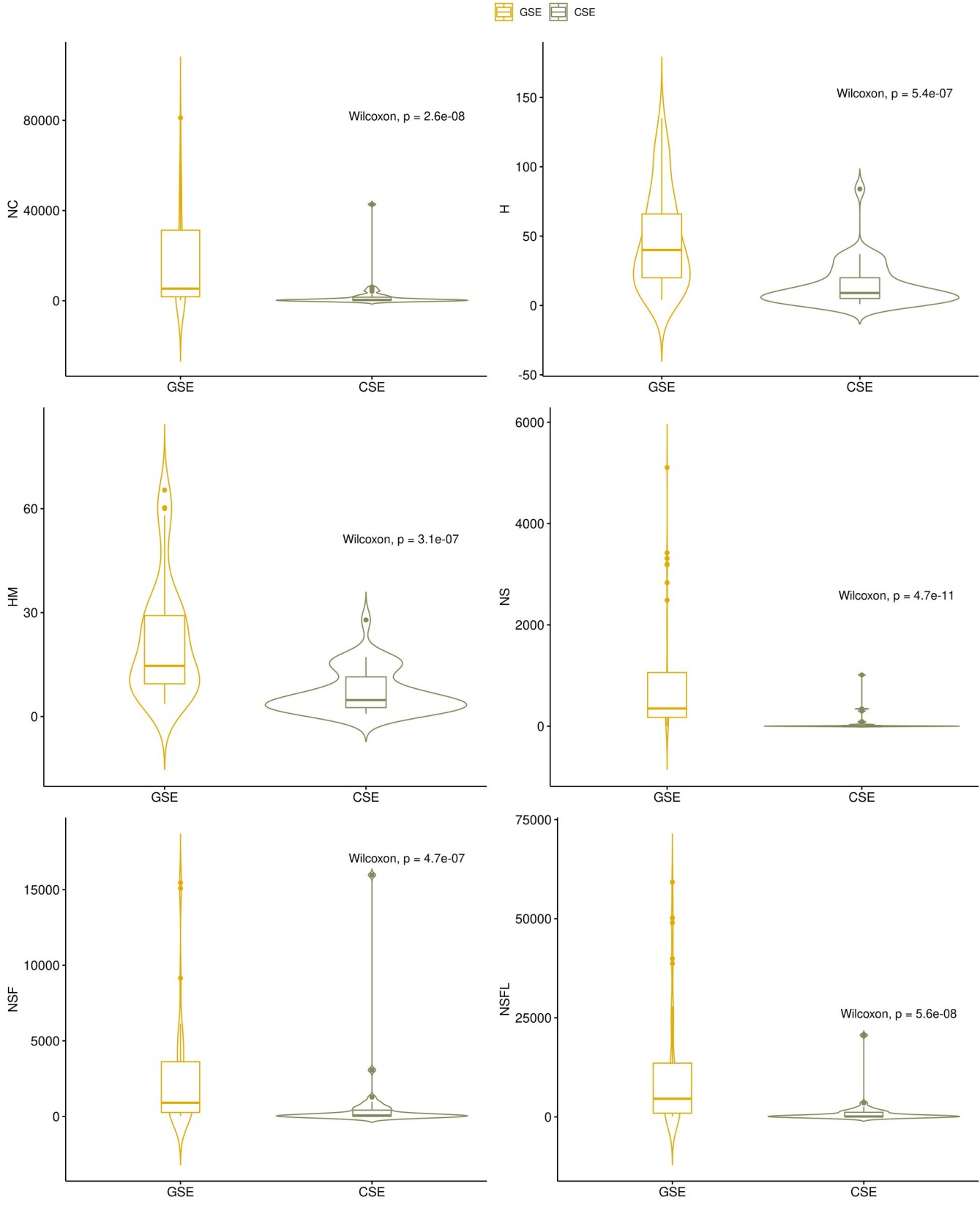

**Fig 15. Box-violin plots according to group of scientific elite and bulk impact and authorship order adjusted impact indices.** Source: the author based on [55, 78, 96]. Note: NC: total citations; H: H index; Hm: Schreiber Hm index; NS: total citations for papers where the scientist is the single author; NSF: total citations for papers where the scientist is the single or first author; NSFL: total citations for papers where the scientist is the single, first, or last author. Source: the authors based on [55, 78, 96].

topics include game theory, a field which was awarded the Nobel Prize in 1994, 1996, 2001, 2005, 2007, 2012 and 2014 [114].

Our findings do not support the post-AAEP *push* effect in citations/article. While a few CSE members received the prize at the peak of their careers (i.e., Muñoz-PhySci; Montes-EnvSci), or after peaks (i.e., Cárdenas-EnvSci), others received the prize far earlier than their most crucial peak (Velez-PhySci). Compared to the GSE—members of which received the Nobel prize at their peak, followed by a brief *halo effect*—there is no discernible *halo effect* for the CSE [22, 115]. A *halo effect* is usually defined as a bias whereby an impression produced by a single trait (i.e., winning a Nobel Prize or an AAEP) influences multiple judgments (i.e., the prize-winning researcher's future research). A contributory factor could be the wide range of participants in the AAEP. Given that the AAEP annually awards diverse products, it is not necessary to be a committed researcher in order to trigger, sustain, or participate in a *halo effect*. This is not the case with the Nobel Prize.

It is also important to mention the lack of sustained and well-funded research in Colombia [42]—a constraint that is likely to persist, resulting in intermittent productivity and impact [116]. National investment in science and technology activities, the leadership of public institutions, and the building of a secure and protected research environment within which ideas can grow and flourish are essential factors that contribute to outstanding research outcomes. For instance, between 2000–2008 the GSE reported a successful record in funding Nobel Prize-winning research by the National Institutes of Health and the National Science Foundation in the US [117]. Even when a substantial fraction of such research is unfunded, public and private research institutions provide a protected and safe environment for brilliant researchers to explore and innovate ideas [117].

Several institutions with GSE members shared rankings with CSE institutions among the top-ten highest betweenness, particularly the PhySci network. For example, institutions such as Harvard University, University of California, Berkeley, and UNAM, showed a higher betweenness in each network. The public-private status of those institutions mirrored the institution networks of the CSE, which were also composed of the local university elite. Some of these universities are private (i.e., Los Andes, Javeriana), some public (i.e., Nacional, Antioquia, Cartagena), but always featuring at the top of the regional rankings [65]. Research centers and agencies in the fields of agriculture and cancer also play an essential role. Thus, despite being high betweenness actors, it is mainly local, GSE-awarded institutions that rank among the top. This does not mean, however, that there is a direct collaboration between the CSE and the GSE–only two researchers had coauthored articles with Nobel laureates. Nevertheless, it supports the idea that Nobel Prize winners influence prize-winning networks and also reflects the *brokerage* role of the CSE as nodes in its respective coauthorship networks [26].

The strategic location of both the GSE and the CSE in the institutional collaboration networks partly reinforces the point made by Jiang and Liu [64]: that top-tier institutions generate the most production and enticement of scientific elites, thereby aggravating the inequality between emergent or peripheral institutions and those with *cumulative advantages*. Furthermore, faculty at high-prestige institutions drive the diffusion and influence of ideas, often irrespective of quality (e.g., an idea spreads more rapidly if it originates from a prestigious institution than an idea of similar quality from a less prestigious institution) [56]. Compared to PhySci and EnvSci, the SoSci institutional/coauthorship networks showed a lower density.

This suggests a closed structure leading to a more efficient flow of knowledge/information, trust and mutual understanding, prosocial group norms, and potential access to support in times of austerity [118]. However, it also comes with downsides such as redundant information and constraints upon actors' options [118]. Conversely, in the open structure of PhySci and EnvSci, new ideas flow through weak ties, and actors with higher betweenness potentially receive strategic resources. Among the downsides, an open structure is not ideal for complex information flow, creates less trust, and complex communication requires more effort [118]. The latter was noticeable in the PhySci author network, where whole clusters were completely exiled from the network.

Macro indices resembled those of previous bibliographic coupling networks modeled to test the Hierarchy of the Sciences hypothesis, particularly for PhySci [119]. The average path length of the PhySci bibliographic coupling network resembled the average path lengths in space science, and physics: 3.6. The EnvSci network showed a lower average path length than the environment/ecology network: 3.7. The consistent average path length in higher consensus (i.e., hardness) disciplines/fields could be explained by references: fewer references are needed to justify/explain/support a study. This observation has to be contrasted with the diversity of disciplinary clusters in PhySci compared to EnvSci since *hardness* in science is characterized by the reduced diversity of sources used (i.e., fewer research topics of general interest). In other words, despite the AAEP being entitled *physics and natural sciences*, the prize has been granted to researchers with a higher diversity in their research fronts—on aggregate—compared to those with a lower diversity in EnvSci and SoSci.

In specific cases, the CSE research fronts differ from Colombia's national output focus. On the other hand, there are clear similarities when comparing the CSE with the main national output disciplines/areas according to ASJC classification. First, while the natural sciences lead national output (mainly: ecology; botany; horticulture; particle physics; and zoology), the PhySci research front composed mainly of medicine was in 4<sup>th</sup> place in the national output ranking. It is also important to bear in mind that the principal component of PhySci has significant involvement in the agricultural and biological sciences, also reflected in the natural science national output. Similarly, in the case of EnvSci—and a substantial portion of the principal component in PhySci—the national output in agricultural sciences figured at the bottom. The second field in the principal component of EnvSci, earth and planetary science, is not significant in the national output. In contrast, the national output in SoSci figured in 2<sup>nd</sup> place (mainly welfare economics; pedagogy; epistemology; law; and social psychology), while the SoSci Scopus profiles were the least represented in the sample. Second, in terms of net output and after applying the ASJC classification to the CSE, the most frequent disciplines/areas (i.e., agricultural and biological sciences; arts and humanities; and medicine) were also among those with the highest output in the country. Other national disciplines/areas by output were engineering and technology (mainly artificial intelligence; food science; control theory; analytical chemistry; and composite material), comprising <1% of the CSE after ASJC classification. This is also consonant with the marginal mathematics cluster in PhySci. Differences in such results need to be viewed with caution since national data was estimated using the Microsoft Academic Graph, which is the second source of bibliographic data in terms of references covered after Google Scholar [120]. Scopus, on the other hand, was in third place in terms of references covered.

Broadening international efforts related to the global development agenda, a comprehensive assessment of SDG using bibliographic coupling identified the following as the most significant clusters: maternal, newborn, and child morbidity and mortality; ecosystems services and adaptations for sustainability; and health surveys, tuberculosis, substance abuse, and longevity [94]. In those areas, the most crowded research fronts of the CSE, PhySci and EnvSci in

particular have the potential and strategic value to contribute to SDG's core research and to participate in its global endeavor [121].

Using bulk impact and authorship adjustment indicators and their composite enabled a more nuanced and inclusive analysis between the CSE and the GSE. Our findings on the first two quartiles of $Ci$ shed light on the local human and scientific profile, seasoned with Nobel productivity/impact features—something not visible either in the pure index of Nobel Prizes affiliated with universities or in Clarivate's Highly Cited Researcher ranking. These changes, inclusions, and exclusions were also pointed out by Ioannidis et al. [55] when proposing the $Ci$ (e.g., Nobel laureates ranked among the top-1,000, but would rank much lower if total citations alone were considered). Finding local names such as Luís Fernando García between William C. Campbell or Riccardo Giacconi puts the knowledge produced by Colombian researchers and developing countries under a very different and encouraging light. The CSE could be found among the superior 50% and the GSE among the inferior 50%. Our findings also contrast with previous findings on the lower research impact of authors from Latin America, despite their involvement as contributors to reputable journals [122].

A final note regarding Zuckerman's pioneering work on the stratification in US Science and the CSE [9, 10]. Zuckerman was particularly concerned about the growing inequality in the distribution of scientific recognition. We found evidence of this in the higher betweenness of institutions with a higher reputation at both local and global level, a reputation, moreover, that is substantially enhanced by having an AAEP awardee affiliated with such institutions. Zuckerman also pointed that—despite such inequality—'no scientist is deprived of all the good things of the scientific life', including those in middle/lower positions [9]. We found this to be reflected in the eclectic and inclusive purpose of the AAEP that goes beyond the '*you must have a PhD to conduct research*' or the 'you must publish—research articles in English in top-tier journals—or perish.' Lastly, Zuckerman lauded science's egalitarian ideology for the way in which it brings together various scientist strata, an interweaving feature and impact in science that emerged in our analysis of $Ci$ results for both the CSE and the GSE.

## Conclusion

This study aimed to provide a comprehensive analysis of the output, impact, and structure of the CSE. It also drew a comparison with the GSE using an impact and normalized authorship composite indicator. Our findings showed that the CSE has a broader agenda than indexed titles in internationally renowned bibliographic databases—mainly local-focused—including PhySci. The CSE showed positive growth compared to the negative growth of the GSE. There were no noticeable changes in productivity/impact among the most prolific researchers before and after receiving the AAEP. CSE-affiliated institutions with the highest betweenness are either local or international reputable institutions—in some cases, multiple Nobel awarded. At the direct coauthorship level, only two researchers published an article with a GSE member. Most of the research profiles reflected the national output priorities, but when research fronts are taken into consideration, the strategic research capacities diverge from the national focus. $Ci$ produced an enriching comparison by showing the scope and reach of Colombian researchers' scientific impact in his/her disciplinary area, even when Nobel laureates are placed in the same assessment framework. The interleaving of the CSE and the GSE—particularly between the 3rd and 2nd quartiles—enabled a more nuanced analysis. Our findings shed light on the research performance-impact standards and agenda between the global North and South and provide an in-context assessment of outstanding local research.

Our study has several limitations. First, the geopolitical scope of the study is limited to Colombia. Second, we did not consider macro-economic variables related to science and

technology activities, such as public/private R&D investment; research grants and scholarships; mentoring; or the correlation between an increased scientific workforce and its scientific impact on the CSE. Such variables could shed light on the potential role of confounding variables. Third, the bibliographic data sourced from Scopus is not a complete picture of the scholarly communication production in Colombia nor in Latin America and the Caribbean.

Further research could source other scientific elites (e.g., *Royal Society Africa Prize*, *Highest Science and Technology Award*, China; *Prêmio Almirante Álavaro Alberto*, Brazil), thereby deepening the understanding of research impact and structure in the context of developing regions/countries. Sourcing bibliographic data from search engines/databases with more comprehensive coverage—such as Google Scholar or Dimensions—could generate further insights by including researchers with no publications in either Scopus or WoS. The inclusion of macroeconomic variables relating to science and technology activities together with the influence of these activities on the impact of the CSE and the national scientific workforce in general could add a socio-economic dimension, thereby yielding a more comprehensive outlook. Lastly, the inclusion of *altmetrics* could provide another perspective: on elites that lie beyond the academic community, such as public debates conducted via social media.

## Supporting information

**S1 Table. Higher/last academic degree by country and university of the CSE.** Source: the authors based on [68] and personal websites.
(XLSX)

## Acknowledgments

The authors thank Dr. Francesca Cauchi for editing an early draft of this manuscript.

## Author Contributions

**Conceptualization:** Julián D. Cortés, Daniel A. Andrade.

**Data curation:** Julián D. Cortés, Daniel A. Andrade.

**Formal analysis:** Julián D. Cortés, Daniel A. Andrade.

**Funding acquisition:** Julián D. Cortés.

**Investigation:** Julián D. Cortés.

**Methodology:** Julián D. Cortés, Daniel A. Andrade.

**Project administration:** Julián D. Cortés.

**Resources:** Julián D. Cortés.

**Software:** Daniel A. Andrade.

**Supervision:** Julián D. Cortés.

**Validation:** Julián D. Cortés, Daniel A. Andrade.

**Visualization:** Julián D. Cortés, Daniel A. Andrade.

**Writing – original draft:** Julián D. Cortés, Daniel A. Andrade.

**Writing – review & editing:** Julián D. Cortés.

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
