## [Decision Letter · Decision Letter 0]

17 Jan 2022

PONE-D-21-30902The Colombian Scientific Elite - Science Mapping and Bibliometric OutlookPLOS ONE

Dear Dr. Cortes,

Thank you for submitting your manuscript to PLOS ONE. After careful consideration, we feel that it has merit but does not fully meet PLOS ONE’s publication criteria as it currently stands. Therefore, we invite you to submit a revised version of the manuscript that addresses the points raised during the review process.

Please, have an especial attention on the recommendations and critiques of referee #1. Minor critiques of referee #3 are also very important to be addressed. Submit your revised manuscript within 40 days. If you will need more time than this to complete your revisions, please reply to this message or contact the journal office at plosone@plos.org. Please include the following items when submitting your revised manuscript:A rebuttal letter that responds to each point raised by the academic editor and reviewer(s). You should upload this letter as a separate file labeled 'Response to Reviewers'.A marked-up copy of your manuscript that highlights changes made to the original version. You should upload this as a separate file labeled 'Revised Manuscript with Track Changes'.An unmarked version of your revised paper without tracked changes. You should upload this as a separate file labeled 'Manuscript'.

We look forward to receiving your revised manuscript.

Kind regards,

Marcelo Hermes-Lima, PhD

Academic Editor

PLOS ONE

Journal Requirements:

"The authors thank Universidad del Rosario’s School of Management and Business for their institutional and financial support. Also, they thank Dr. Francesca Cauchi for editing an early draft of this manuscript."

"JDC

Seed fund

Universidad del Rosario

https://www.urosario.edu.co/

Reviewers' comments:

Reviewer's Responses to Questions

**Comments to the Author**

1. Is the manuscript technically sound, and do the data support the conclusions?

Reviewer #1: Partly

Reviewer #2: Yes

Reviewer #3: Yes

2. Has the statistical analysis been performed appropriately and rigorously? 

Reviewer #1: I Don't Know

Reviewer #2: Yes

Reviewer #3: N/A

3. Have the authors made all data underlying the findings in their manuscript fully available?

Reviewer #1: Yes

Reviewer #2: Yes

Reviewer #3: No

4. Is the manuscript presented in an intelligible fashion and written in standard English?

Reviewer #1: Yes

Reviewer #2: Yes

Reviewer #3: Yes

5. Review Comments to the Author

Reviewer #1: The manuscript The Colombian Scientific Elite - Science Mapping and Bibliometric Outlook is an analysis of bibliometric tools applied to the scientific production of Colombian scientists that were awarded with the Alejandro Ángel Escobar national prize in the timeframe of 1990-2020. The bibliometric tools are used to establish a correlation between the carrier and production of the top scientists with the parameters that could correlate to their chances of winning a prize. Parallels with other world prizes such as the Nobel prize are carried to evaluate if the Colombian award has been fairly attributed. The authors base their evaluation models of impact and productivity on the variables of density and average path length, community detection, and betweenness. Besides, they consider other variables such as H_index, Hm_index, effective rank. Although the manuscript contains a good discussion regarding the relevance of the work of laureate scientists, the form they chose to present their data does not allow the reader to establish direct relations between their claims and the given data.

For those reasons, I recommend a major thorough revision before its acceptance.

Here I list some of the points that need to be clarified/corrected:

1) For instance, they use a confusing nomenclature in Table 3, page 17 out of 44. There is no direct correlation between column names and the mentioned variables in the methodology section a few lines and pages above. It seems that they present confusing data with wrong words. For example, there are no Authors per article less than one! This name is mistaken (the English use too since it is Authors per article and not Authors per articles).

2) Other English errors/mistranslated concepts are all over the manuscript, such as coauthorship adjustment that would be a normalized coauthorship. An adjustment would be a fitting (a better word to be used), but it does not contain the correct meaning.

3) It is unclear where to classify all the presented parameters as Macro, meso, or micro. B_coup, hm, r_eff, authors per article? Which category to organize them? And why are those variables defined if not all of them are in a single table for comparison?

4) There are also problems with graph presentation that uses lines to connect points that are uncorrelated through time; a large number of years that difficult reading them in an axis; comparison between large numbers and smaller ones that become flat in a scale should be made in a logarithm scale.

5) There are issues with the wrong choice of areas. For instance, Natural Sciences are such a massive cluster and have distinct areas that are too difficult to compare. Medical sciences with Physics cannot be a good choice of comparison. It is like comparing two countries, one with a population ten times higher than the other. The choice of the Award academy of area group classification is no excuse for carrying the analysis of those areas together.

6) References have some issues to I identified that reference 26 is repeated in 99. The is a ? in reference 98 and some internet sites that are not accessed recently. Authors should verify if they're still active at the time of the submission.

Reviewer #2: An important objective of the present study, focusing on the Scientific Elite of Colombia, is to understand the performance and structure of research in developing countries and the existing levels of cooperation compared to the Global Scientific Elite. The study is quite original in presenting an analysis methodology that allows the comparison of the scope and reach of scientific production and the respective impacts of elite groups in a country in relation to Nobel laureates using the same evaluation structure.

Reviewer #3: The authors carried out a comprehensive analysis of the output, impact, and structure of “the Colombian scientific elite”, which is composed of researchers awarded with the Alejandro Ángel Escobar Foundation National Prize - the Colombian Nobel. Their analysis indicated that the Colombian scientific elite had a “positive growth” and that there was an inverse trend compared to that for Nobel Prize laureates’ productivity. “There were no noticeable changes in productivity/impact before and after receiving the prize.”. The authors also show that Colombian scientific elite members are not active partners in collaborative networks of Nobel Prize laureates. In their mapping and bibliometric outlook, they drew upon coauthorship networks (institutional and authorial), bibliographic coupling, and a sample of 87 awardees. It is interesting to note that their analysis included social sciences and humanities, which is not common in other studies.

The work is timely, and the results shed light on the traits of the “scientific elite” in Latin American countries. To me, the manuscript adds to the understanding of factors underlying Colombian science and to the social studies of science at large.

I have minor points that I would like the authors to address:

1. On page 3, l.82, the authors mention that the limitations of the study and future agenda would be presented in the end. Not that clear to me. I did not identify details on limitations.

2. On page 3, l.78-80, the authors write that “The methods and techniques implemented are coauthorship 79 networks, both at the institutional and author levels; bibliographic coupling; and a comparative sample of 82 researchers using the composite indicator proposed by Ioannidis et al. [52]”. Then, they describe a sample of 87 awardees. Please clarify.

3. On page 29, l. 570-571 the authors state that “Compared to the GSE ─ members of which received the Nobel prize at their peak, followed by a brief halo effect ─ there is no discernible halo effect for the CSE [22,106].”. [L. 576-578] They write that “It is also important to mention the lack of sustained and well-funded research in Colombia [107] ─ a constraint that is likely to persist, resulting in intermittent productivity and impact [108].” I think this lack of sustained funding makes this analysis a little bit complicated – especially because this is not usually the case for the countries of most Nobel prize winners. Please explain why your comparison to the GSE would be reliable despite this fator.

4. As Argentina has a marked role as a leading Latin American country in terms of Nobel prizes (https://eserb.cancilleria.gob.ar/en/2021-year-tribute-nobel-prize-medicine-dr-c%C3%A9sar-milstein), I missed a broader context for the discussion of the results. Please justify why Argentina is not mentioned in your discussion.

5. For readers not familiar with scientific indicators for Colombia, it’s difficult to figure out the whole picture offered by this study. Data on GDP in research and development, number of active scientists, and others are missing.

6. There are no comments on possible counfounders in the whole analysis. Could you please comment on them?

7. R. K. Merton and H. Zuckerman’s contributions could be more evident in the rationale of the study and discussion of results. They are cited but do not have an active part in the discussion for this particular Latin America scenario. Does the concept of scientific elite put forward by Zuckerman apply to the Colombian scientific elite?

8. Figures and Tables should be self-contained, as we know. Please revise those which are not. Example: “Table 6 Composite index (C) for the CSE and GSE” (page.25, l.519)

6. PLOS authors have the option to publish the peer review history of their article (what does this mean?). If published, this will include your full peer review and any attached files.

Reviewer #1: **Yes: **Rero Marques Rubinger

Reviewer #2: **Yes: **BENEDITO GUIMARÃES AGUIAR NETO

Reviewer #3: No

---

## [Author Response · Author response to Decision Letter 0]

31 Jan 2022

Dear reviewers—

I have submitted a 'letter of response' as an attached document in the submission. Also, the manuscript with/without track changes. 

Sincerely, 

The authors

---

## [Decision Letter · Decision Letter 1]

18 Mar 2022

PONE-D-21-30902R1The Colombian scientific elite — Science mapping and a comparison with Nobel Prize laureates using a composite citation indicatorPLOS ONE

Dear Dr. Cortes,

Thank you for submitting your manuscript to PLOS ONE. After careful consideration, we feel that it has merit but does not fully meet PLOS ONE’s publication criteria as it currently stands. Therefore, we invite you to submit a revised version of the manuscript that addresses the points raised during the review process. Referee 1 still has requested minor changes. The other referees are OK with your work.

Please submit your revised manuscript within 20 days. If you will need more time than this to complete your revisions, please reply to this message or contact the journal office at plosone@plos.org. Please include the following items when submitting your revised manuscript:A rebuttal letter that responds to each point raised by the academic editor and reviewer(s). You should upload this letter as a separate file labeled 'Response to Reviewers'.A marked-up copy of your manuscript that highlights changes made to the original version. You should upload this as a separate file labeled 'Revised Manuscript with Track Changes'.An unmarked version of your revised paper without tracked changes. You should upload this as a separate file labeled 'Manuscript'.

We look forward to receiving your revised manuscript.

Kind regards,

Marcelo Hermes-Lima, PhD

Academic Editor

PLOS ONE

Journal Requirements:

Reviewers' comments:

Reviewer's Responses to Questions

**Comments to the Author**

1. If the authors have adequately addressed your comments raised in a previous round of review and you feel that this manuscript is now acceptable for publication, you may indicate that here to bypass the “Comments to the Author” section, enter your conflict of interest statement in the “Confidential to Editor” section, and submit your "Accept" recommendation.

Reviewer #1: All comments have been addressed

Reviewer #2: All comments have been addressed

Reviewer #3: All comments have been addressed

2. Is the manuscript technically sound, and do the data support the conclusions?

Reviewer #1: Yes

Reviewer #2: Yes

Reviewer #3: Yes

3. Has the statistical analysis been performed appropriately and rigorously? 

Reviewer #1: Yes

Reviewer #2: Yes

Reviewer #3: (No Response)

4. Have the authors made all data underlying the findings in their manuscript fully available?

Reviewer #1: Yes

Reviewer #2: Yes

Reviewer #3: Yes

5. Is the manuscript presented in an intelligible fashion and written in standard English?

Reviewer #1: Yes

Reviewer #2: Yes

Reviewer #3: Yes

6. Review Comments to the Author

Reviewer #1: The manuscript describes bibliometric tools to carry on a comparative analysis between Nobel Laureates and Colombian Alejandro Ángel EscobarFoundation National Prize Laureates from 1990 to 2020. For this, they considered The following parameters: Coauthorship networks map of scientific collaboration; Composite citation indicator, normalized coauthorship; and author order based on six indicators at the author level; H and Hm indexes among other indexes. The manuscript has improved substantially compared with the previous version but it still needs a few minor issues to be addressed before its acceptance that I list in the following:

1) The abstract needs a full revision. It is not written as an abstract that should address what was done and the main conclusions drawn from the results in short.

2) Rewrite the sentence on page 9 line 63: “Colombia does, however, have its own Nobel Prize:” it is not a “Nobel prize” since the prize is affiliated to a distinct institution and may have other forms of choosing their winners. Something like a Nobel equivalent should be more adequate.

3) Page 65 Table 3 first line column H index is written 1,03. Is it the Hm index ou maybe 1030? Please correct.

Reviewer #2: The work presents the necessary academic solidity and, in my point of view, the authors responded to the suggestions made by the other reviewers.

Reviewer #3: Thank you for the detailed responses. My additional concerns were addressed, and I have no further comments. I'm happy to recommend acceptance.

7. PLOS authors have the option to publish the peer review history of their article (what does this mean?). If published, this will include your full peer review and any attached files.

Reviewer #1: No

Reviewer #2: **Yes: **Aguiar Neto, B. G.

Reviewer #3: No

---

## [Author Response · Author response to Decision Letter 1]

22 Mar 2022

Dear reviewers—

I have submitted a 'letter of response' as an attached document in the submission. Also, the manuscript with/without track changes. 

Sincerely, 

The authors

---

## [Decision Letter · Decision Letter 2]

16 May 2022

The Colombian scientific elite — Science mapping and a comparison with Nobel Prize laureates using a composite citation indicator

PONE-D-21-30902R2

Dear Dr. Cortes,

We’re pleased to inform you that your manuscript has been judged scientifically suitable for publication and will be formally accepted for publication once it meets all outstanding technical requirements.

Kind regards,

Marcelo Hermes-Lima, PhD

Academic Editor

PLOS ONE

Additional Editor Comments (optional):

Reviewers' comments:

Reviewer's Responses to Questions

**Comments to the Author**

1. If the authors have adequately addressed your comments raised in a previous round of review and you feel that this manuscript is now acceptable for publication, you may indicate that here to bypass the “Comments to the Author” section, enter your conflict of interest statement in the “Confidential to Editor” section, and submit your "Accept" recommendation.

Reviewer #1: All comments have been addressed

2. Is the manuscript technically sound, and do the data support the conclusions?

Reviewer #1: Yes

3. Has the statistical analysis been performed appropriately and rigorously? 

Reviewer #1: I Don't Know

4. Have the authors made all data underlying the findings in their manuscript fully available?

Reviewer #1: Yes

5. Is the manuscript presented in an intelligible fashion and written in standard English?

Reviewer #1: Yes

6. Review Comments to the Author

Reviewer #1: The Authors have revised the manuscript according to the issues addressed in the previous review analysis.

The manuscript is suitable to be published in the current form

7. PLOS authors have the option to publish the peer review history of their article (what does this mean?). If published, this will include your full peer review and any attached files.

Reviewer #1: **Yes: **Rero Marques Rubinger

---

## [Editor Report · Acceptance letter]

18 May 2022

PONE-D-21-30902R2 

The Colombian scientific elite — Science mapping and a comparison with Nobel Prize laureates using a composite citation indicator 

Dear Dr. Cortés:

I'm pleased to inform you that your manuscript has been deemed suitable for publication in PLOS ONE. Congratulations! Your manuscript is now with our production department. 

Kind regards, 

on behalf of

Dr. Marcelo Hermes-Lima 

Academic Editor

PLOS ONE